# Selective sweep for an enhancer involucrin allele identifies skin barrier adaptation out of Africa

Mary Elizabeth Mathyer [1,2,3,6], Erin A. Brettmann [1,2,3,6], Alina D. Schmidt[1,2,3], Zane A. Goodwin[1,2,3], Inez Y. Oh[1,2,3], Ashley M. Quiggle[1,2,3], Eric Tycksen[4], Natasha Ramakrishnan[1,2,3], Scot J. Matkovich[2], Emma Guttman-Yassky [5], John R. Edwards[2] & Cristina de Guzman Strong [1,2,3✉]

The genetic modules that contribute to human evolution are poorly understood. Here we investigate positive selection in the Epidermal Differentiation Complex locus for skin barrier adaptation in diverse HapMap human populations (CEU, JPT/CHB, and YRI). Using Composite of Multiple Signals and iSAFE, we identify selective sweeps for *LCE1A-SMCP* and involucrin (*IVL*) haplotypes associated with human migration out-of-Africa, reaching near fixation in European populations. CEU-*IVL* is associated with increased *IVL* expression and a known epidermis-specific enhancer. CRISPR/Cas9 deletion of the orthologous mouse enhancer in vivo reveals a functional requirement for the enhancer to regulate *Ivl* expression in *cis*. Reporter assays confirm increased regulatory and additive enhancer effects of CEU-specific polymorphisms identified at predicted IRF1 and NFIC binding sites in the *IVL* enhancer (rs4845327) and its promoter (rs1854779). Together, our results identify a selective sweep for a *cis* regulatory module for CEU-*IVL*, highlighting human skin barrier evolution for increased *IVL* expression out-of-Africa.

[1] Division of Dermatology, Department of Medicine, Washington University in St. Louis School of Medicine, St. Louis, MO 63110, USA. [2] Center for Pharmacogenomics, Department of Medicine, Washington University in St. Louis School of Medicine, St. Louis, MO 63110, USA. [3] Center for the Study of Itch & Sensory Disorders, Washington University in St. Louis School of Medicine, St. Louis, MO 63110, USA. [4] McDonnell Genome Institute, Washington University in St. Louis, St. Louis, MO 63110, USA. [5] Department of Dermatology, Icahn School of Medicine at Mt. Sinai, New York, NY 10029, USA. [6] These authors contributed equally: Mary Elizabeth Mathyer, Erin A. Brettmann ✉email: cristinastrong@wustl.edu

Modern humans (*Homo sapiens*) have evolved by adapting to local environments and niches, under constant pressure to survive[1]. Our current understanding of human evolution has been founded on visual phenotypic changes and the underlying genetic variation. However, whole-genome sequencing and downstream methodology have facilitated a reverse genomics approach, allowing researchers to more accurately pinpoint significant genetic differences and thus reveal additional adaptive traits[2,3]. In particular, the availability of multiple and diverse human genomes has revolutionized the field, enabling the identification of loci undergoing selection in different populations and their impact on human health[3,4]. Here we investigate human skin barrier adaptation by examining signatures of positive selection within the epidermal differentiation complex (EDC) locus from a diverse set of human genomes. The EDC (located on human 1q21, mouse 3q) exhibited the highest rate of non-synonymous substitutions in the human genome and was ranked as the most rapidly diverging gene cluster in the human–chimp genome comparison[5]. The EDC spans approximately 1.6 Mb and contains 64 genes representing four gene families, including the filaggrin (FLG)-like or SFTP (S100-fused-type protein), late cornified envelope (LCE), small proline repeat-rich (SPRR), and S100-domain (S100) family members[6,7]. The expression of many of these EDC genes is a hallmark feature of the terminally differentiated epidermal cells (keratinocytes) that comprise the interfollicular epidermis and form the first line of defense at the skin surface[8]. Many EDC proteins, including involucrin (IVL) and many SPRRs and LCEs, are covalently cross-linked to form the cornified envelope that surrounds the keratinocyte[8]. Identification of EDC orthologs across many mammalian genomes[9–12] led to the discovery of the observed linearity and synteny of the EDC in both mammals and vertebrates[6,713–17]. We previously identified positive selection in the EDC for specific genes across the mammalian phylogeny, and specifically in primates and humans[18]. Motivated by these studies, we hypothesized ongoing evolution within the EDC for skin barrier adaptation among the geographically diverse group of modern-day human populations.

Here, we show selective sweeps for *LCE1A-SMCP* and *IVL* haplotypes associated with human migration out-of-Africa. The European CEU-*IVL* haplotype is associated with increased *IVL* expression and includes an epidermis-specific enhancer. Deletion of the enhancer in mice using CRISPR/Cas9 genome editing results in a significant decrease in *Ivl* expression in *cis*. We translate this new knowledge of the enhancer to regulate *IVL* in humans and examine human population-specific alleles for *IVL*. We find increased regulatory activity specific to CEU-*IVL* with an additive effect by the enhancer that includes predicted binding for IRF1 at enhancer rs4845327 and NFIC binding at promoter rs1854779. Together, the findings highlight recent human skin barrier evolution for increased IVL expression and its enhancer, as humans migrated out-of-Africa.

## Results

### Selective sweeps for the CEU-*LCE1A*-*SMCP* and CEU-*IVL* haplotypes out-of-Africa

We sought to determine positive selection in the EDC using two independent algorithms, composite of multiple signals (CMS)[19,20] and the integrated selection of allele favored by evolution (iSAFE)[21] (Fig. 1a). CMS comprehensively identifies sites that are most likely to have undergone a positive selective sweep, reporting a composite probability score from multiple selection tests for a given SNP[19]. iSAFE incorporates coalescent structures surrounding regions under selective sweep to further rank and pinpoint sites of positive selection[21]. CMS scores for all EDC HapMap II SNPs were extracted for each

of the following populations: individuals from (1) Utah of European descent (CEU), (2) Yoruba from Ibadan, Nigeria (YRI), and pooled Japanese in Tokyo, Japan and Han Chinese in Beijing (JPT/CHB)[22]. Signals of positive selection using CMS were not detected in YRI (false discovery rate, FDR < 0.05, Supplementary Data 1). However, rs4511111 near *HRNR* was positively selected in JPT/CHB (Fig. 1b) and is in strong linkage disequilibrium (LD) ($r^2 \geq 0.8$) with the positively selected *HRNR-FLG* "Huxian haplogroup" previously reported in the Han Chinese population[23] which further validates the significance of the Huxian haplogroup sweep. By contrast, evidence of positive selection in CEU was found in two genomic regions: rs3007674 nearby *S100A11* and a cluster of four SNPs near *SMCP* (rs12022319, rs4845490, rs4845491, and rs3737861) that collectively exhibited the strongest signal in CEU ($3.64 \leq CMS_{GN} \leq 4.75$; $CMS_{GN}$, genome-normalized composite of multiple signals). Thus, our findings using CMS identifies selective sweeps for *HRNR-FLG* in JPT/CHB and near *S100A11* and *SMCP* in CEU.

We sought to validate and refine the positively selected regions identified by CMS using iSAFE[21]. iSAFE incorporates the "shoulders" that are proximal to the region under selective sweep and ranks the signals to further identify the favored mutation. iSAFE scores were calculated for 1000 Genomes Project (1KGP) Phase 3 SNPs within the EDC in YRI, JPT/CHB, and CEU populations, with evidence of positive selection defined as an iSAFE score greater than 0.1 (empirical $p < 1 \times 10^{-4}$)[21]. We found no evidence of positive selection in YRI using iSAFE consistent with the CMS negative findings (Supplementary Data 2). However, iSAFE revealed positive selection in JPT/CHB between *LCE1F* and *LCE1B* (rs10157301, rs1930127, and rs11804609), but did not replicate the CMS finding for *HRNR-FLG* (Fig. 1c and Supplementary Data 2). iSAFE also identified three genomic regions under positive selection in CEU. Signals for the *HRNR-CRNN* and *LCE2D* regions were newly discovered (Fig. 1c and Supplementary Data 2). The *HRNR-CRNN* signals span multiple genes in the S100-fused family, including *HRNR, FLG, FLG2, CRNN*, and the noncoding RNA *FLG-AS1*. More importantly, iSAFE validated the same *SMCP* region identified by CMS with evidence of a relatively broader region under selection (Fig. 1c). The same four CMS SNPs near *SMCP* were among a cluster of iSAFE signals (13 SNPs) between *LCE1B, LCE1A*, and *LCE6A*. To the right of this *LCE1B-SMCP* region, a shoulder of iSAFE signals (7 SNPs) upstream and within *IVL* was also newly detected with scores at 0.097. Similar signals were not observed in the shoulders of other positively selected regions, suggesting either a strong hitchhiking effect of this shoulder or perhaps a very recent driver for the positively selected *LCE1B-SMCP* region. iSAFE signals (0.095) in CEU were also detected near *SPRR1B* but were isolated indicative of a relatively soft selective sweep. Thus, our iSAFE results reveal additional findings for selective sweeps near *LCE1F* and *LCE1B* in JPT/CHB and *HRNR-CRNN* and *LCE2D* in CEU. iSAFE provided higher resolution and further validation of CMS signal surrounding the *LCE1B-SMCP* region and elucidation of the positively selected *IVL* shouldering region in CEU that we also found to be positively selected in European 1KGP populations, FIN (Finnish in Finland) and IBS (Iberian population in Spain) (Supplementary Data 2 and Supplementary Fig. 1). Together, both CMS and iSAFE findings identify population-specific signatures of positive selection in the EDC with notable selective sweeps that span the *LCE1B-IVL* region, highlighting human evolution associated with the skin barrier.

We next determined the haplotype structure(s) for the positively selected signals found in the *LCE1B-SMCP* and *IVL* regions in CEU (Supplementary Data 3). The *LCE1B-SMCP* region is marked by a 65.2 kb haplotype, referred to as CEU-*LCE1A-SMCP*, whereas the *IVL* region is marked by 41.2 kb haplotype including the *IVL* gene,

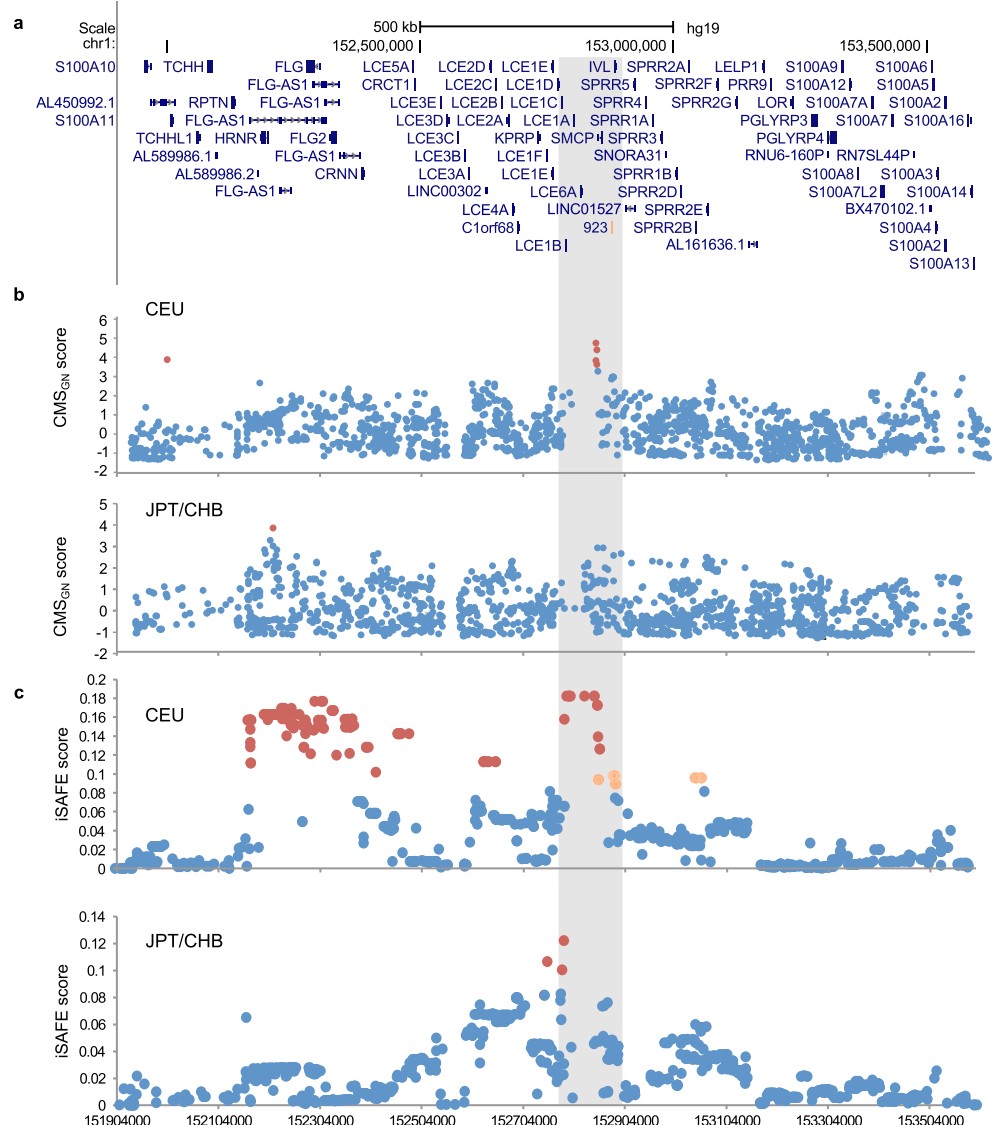

**Fig. 1 Human evolution in the skin barrier as evidenced by multiple population-specific signals for positive selection in the epidermal differentiation complex. a** Positive selection within the EDC (hg19 [human genome]; chr1:151,904,000-153,593,700) was determined by: **b** clusters of SNPs with genome-normalized (GN) composite of multiple signals ($CMS_{GN}$) score > 2 with a FDR < 0.05, and **c** SNPs with iSAFE scores > 0.10, all shown as red dots. Orange dots indicate SNPs 0.095<iSAFE score < 0.10. The gray shaded region indicates the region of shared evidence of positive selection in CEU (Utah of European descent) near *LCE1B-IVL*. CMS scores were previously calculated for 1KGP HapMap II SNPs in CEU and JPT/CHB (Japanese in Tokyo, Japan and Han Chinese in Beijing) populations and iSAFE scores were calculated for 1KGP Phase 3 SNPs in CEU and JPT/CHB. Here, 923 enhancer is shown as orange bar in panel (**a**). kb, kilobase.

referred to as CEU-*IVL* (Fig. 2a). The same haplotypes were also found in FIN and IBS (Supplementary Data 3). As the SNPs found in the CEU-*LCE1A-SMCP* and CEU-*IVL* haplotypes are globally distributed, we hypothesized recent selective sweeps for CEU-*LCE1A-SMCP* and CEU-*IVL* that correlate with an out-of-Africa migration. To test this hypothesis, we examined the relationships between the allele frequencies for the CEU-*LCE1A-SMCP* (rs4845490-A) and CEU-*IVL* (rs4845327-G) haplotypes in a given 1KGP population to the population's latitude used as a proxy for human migration. We found a direct and positive correlation for the SNP frequency to northern latitude (rs4845490-A, $R^2 = 0.88$, $\rho = 1.7 \times 10^{-5}$; rs4845327-G, $R^2 = 0.88$, $\rho = 2.2 \times 10^{-5}$; Fig. 2b) with near-fixation allele frequency (0.96) observed in FIN. These findings are in contrast to a globally common EDC SNP, rs2711, which was not positively correlated ($R^2 = 0.55$, $\rho = 8.6 \times 10^{-3}$, Supplementary Fig. 2). Together, these data reveal selective sweeps for the CEU-

*LCE1A-SMCP* and CEU-*IVL* haplotypes in European populations that are associated with migration out-of-Africa.

**CEU-*IVL* haplotype is associated with increased *IVL* gene expression.** We next determined the functional significance of the CMS/iSAFE SNPs for the positively selected CEU-*LCE1A-SMCP* and CEU-*IVL* haplotypes using the Genotype-Tissue Expression (GTEx) project[24]. Of the four CMS/iSAFE SNPs on the CEU-*LCE1A-SMCP* haplotype, only rs4845491 and rs16834728 were identified as expression quantitative trait loci (eQTLs) that were specific to sun-exposed skin. Their T alleles for CEU-*LCE1A-SMCP* were associated with increased expression for *HRNR*, a gene located outside of the haplotype (Supplementary Fig. 3). By contrast, all seven iSAFE SNPs for the CEU-*IVL* haplotype were annotated as eQTLs associated with increased *IVL* expression in both sun-

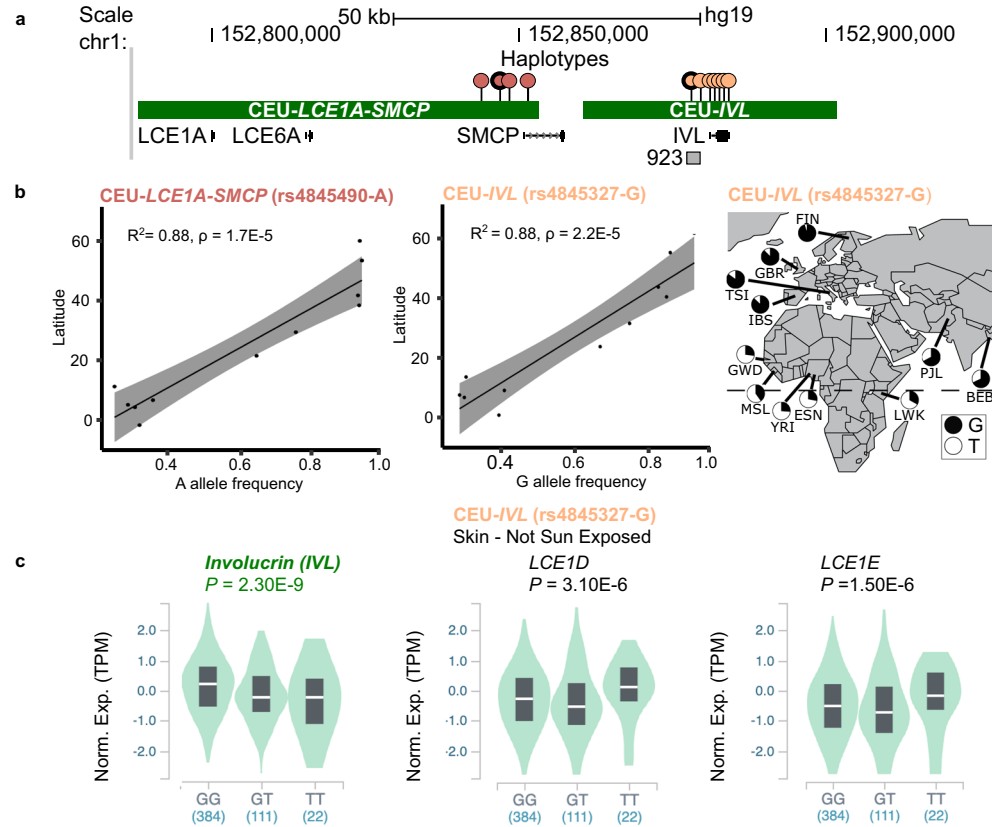

**Fig. 2 Selective sweeps for CEU-*LCE1A-SMCP* and CEU-*IVL* out-of-Africa. a** Phasing reveals CEU-*LCE1A-SMCP* and CEU-*IVL* haplotypes (green bars) in CEU based on SNPs in linkage disequilibrium ($r^2 > 0.8$) with the high CMS/iSAFE (red; rs12022319, rs4845490 [bolded], rs4845491, & rs3737861) and iSAFE (orange; rs4845327 [bolded], rs1854779, rs7539232, rs11205132, rs2229496, rs7535306, & rs7545520) SNPs (lollipops). CEU-*IVL* includes *IVL* and the epidermis-specific 923 enhancer, whereas CEU-*LCE1A-SMCP* includes *LCE1A*, *LCE6A*, and *SMCP*. **b** Direct and positive correlations between the frequencies of rs4845490-A & rs4845327-G, and Northern latitudes reveal associations of the CEU-*LCE1A-SMCP* and CEU-*IVL* selective sweeps with out-of-Africa migration ($\rho = 1.7$E-5 and 2.2E-5), respectively. Black line indicates a linear relationship (Pearson's correlation) between allele frequency and geographic latitude, two-sided *t*-test (9 degrees of freedom) with gray area surrounding the regression line representing 95% confidence intervals for the group mean values of the latitude for each allele frequency. Pie charts of rs4845327-G allele frequency for each population are indicated on the map. **c** Violin plots for rs4845327-G for CEU-*IVL*, an eQTL for increased *IVL* and decreased *LCE1D* and *LCE1E* expressions shown in not sun-exposed skin (GTEx [V8]). Box in violin plot represents interquartile range with median (white line). Numbers in parenthesis indicate number of individuals for each genotype. Norm. Exp., normalized expression; TPM, transcripts per million. Chi-Square *p*-value was calculated based on Mahalanobis distance and Bonferroni corrected.

exposed and not sun-exposed skin (Fig. 2c, representative SNP rs4845490-G shown; and Supplementary Table 1). The SNPs were also eQTLs for decreased expression of *LCE1E* and *LCE1D*, genes located outside the CEU-*IVL* haplotype. The findings for differential *IVL* expression suggest a role for enhancer regulation with population-specific variation. We previously identified and characterized a strong epidermis-specific enhancer located upstream of *IVL*[7] (Figs. 2a and 3a). This enhancer, termed "923" is located 923 kb from the most centromeric EDC gene *S100A10* and is associated with the dynamic chromatin remodeling of the EDC and cJun/AP1 binding concomitant with EDC gene expression[25]. The CEU-*IVL* haplotype contains the 923 enhancer and *IVL*. This led us to hypothesize that the 923 enhancer modulates *IVL* expression associated with positive selection.

**Deletion of the 923 enhancer in mice results in decreased expression of the proximal gene involucrin in the epidermis.**
To determine the function of the 923 enhancer, we generated knockout mice for the orthologous 923 enhancer using CRISPR/Cas9 genome editing (Fig. 3a). Our strategy aimed to delete the conserved noncoding element 923 with known epidermis-specific

regulatory activity[7,25] and epigenetically marked by H3K27ac histone acetylation observed in ENCODE human keratinocytes. We targeted a larger 923 enhancer region given the H3K27ac epigenetic mark. Two independent deletions were successfully generated, resulting in a specific deletion of the 923 enhancer (923[del]) and a large 40 kb deletion that included the 923 enhancer and proximal genes *Smcp* and *2210017I01Rik* (923[large]) (Fig. 3a, b). At birth, 923[del] and 923[large] C57BL/6 (B6) knockout mice were viable and did not deviate from the expected genotype ratios for both heterozygous parental crosses ($X^2$ test, $\alpha = 0.05$) (Supplementary Table 2). We also observed no morphological differences or defects in barrier function under homeostatic conditions (Supplementary Fig. 4). We next examined the molecular impact of the 923 enhancer deletion on the skin transcriptome. RNA-seq on 923[del] and 923[large] homozygous, heterozygous, and wild-type (WT) newborn skins was performed (Supplementary Tables 3 and 4, and Supplementary Data 4 and 5). Given the close proximity of the 923 enhancer to involucrin (*Ivl*), we included the *Ivl* B6 knockout mouse[26] in our analyses to distinguish the direct effect of the loss of the enhancer from the hypothesized loss of Ivl (Supplementary Data 6). Analysis of the skin transcriptomes identified 6 significantly differentially

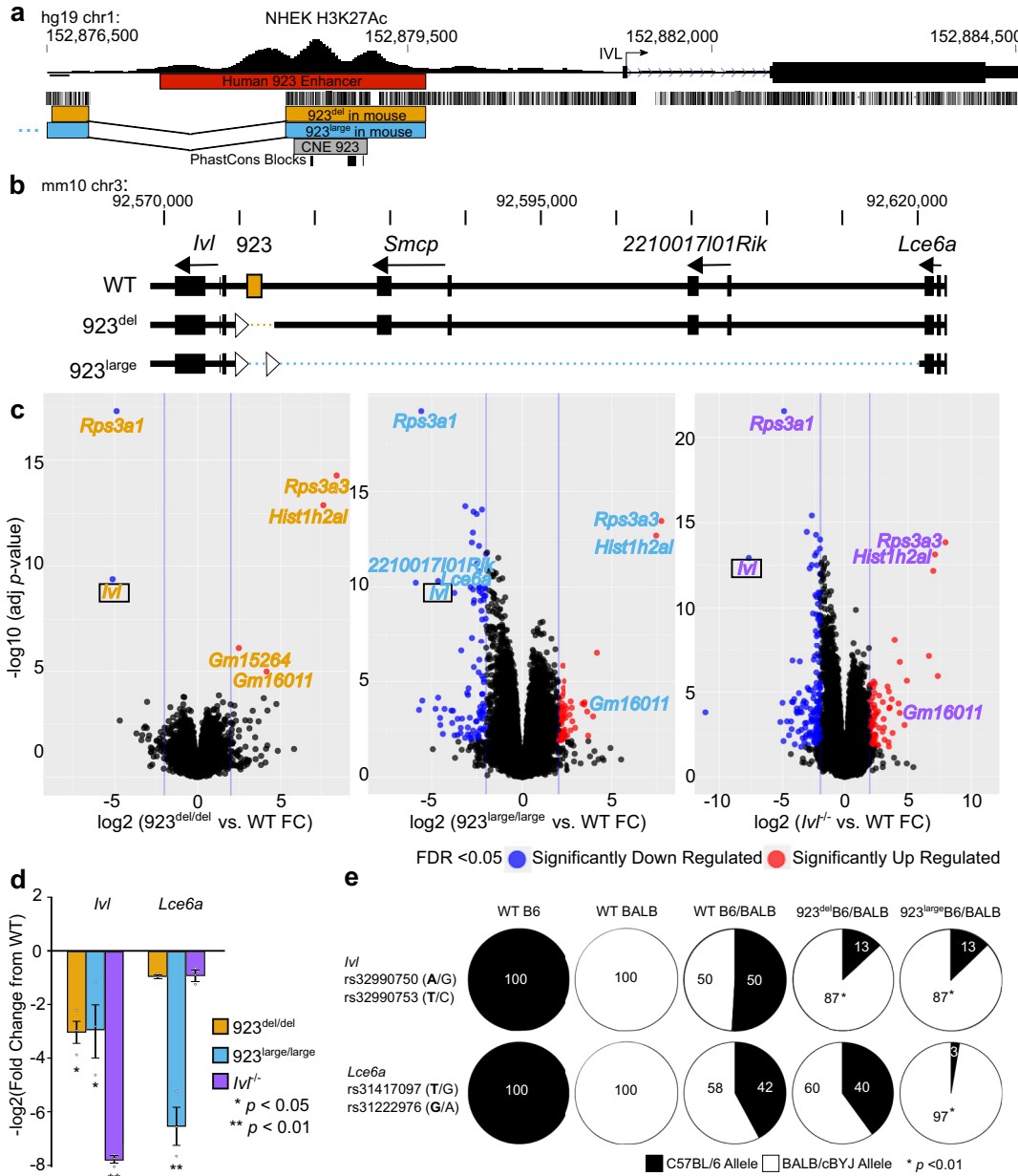

**Fig. 3 Deletion of the 923 enhancer *in vivo* identifies involucrin as a target gene via *cis* regulation. a** Schematic of the human 923 enhancer (red) as defined by H3K27Ac marks in NHEK (Normal Human Epidermal Keratinocytes); hg19; chr1:152,877,349-152,879,610, the conserved noncoding element (CNE) 923 with PhastCons blocks (gray); hg19; chr1:152,878,419-152,879,075 and the orthologous deleted enhancer sequence (923[del]; orange) and larger deletion (923[large]; blue) in the mouse with alignments to human as shown. **b** Generation of two 923 enhancer deletion alleles by CRISPR/Cas9 genome editing in mice shown as dotted lines, 923[del] (mm10 [*Mus musculus*]; chr3: 92,575,843 – 92,577,094): Deletion of mouse orthologous 923 enhancer; 923[large]: ~40 kb deletion (mm10; chr3: 90,575,863 – 92,577,075) that includes 923 enhancer, *Smcp*, and *2210017I01Rik* annotations. Intact flanking *loxP* sites (triangles) were also successfully introduced via homologous recombination. **c** Volcano plots of ribosomal-zero RNA-seq differential gene expression analyses of 923[del/del] (orange text), 923[large/large] (light blue text), and *Ivl*[−/−] (purple text) newborn mouse skins each compared to wild-type (WT) reveal significant decreased expressions for *Ivl*. *n* = 3/genotype. FC, fold change. Statistical analysis using Limma's generalized linear model moderated two-sided *t*-tests with 22 degrees of freedom and Benjamini-Hochberg false discovery rate (FDR) corrections. Significant data points with FDR ≤ 0.05 in blue are log2 fold down-regulated ≤ −2 and red are up-regulated ≥2. **d** Confirmatory qPCR validation for decreased *Ivl* expression for 923[del/del] (*p* = 0.030), 923[large/large] (*p* = 0.034), and *Ivl*[−/−] (*p* = 0.001), and additional decreased *Lce6a* expression specific to 923[large/large] newborn mouse skins (*p* = 0.001) all compared to WT mice; *n* = 3/genotype; Error bars ±SEM. One-way ANOVA followed by Tukey's HSD displayed. **e** Proportions of allele-specific (SNP) *Ivl* and *Lce6a* expression shown in pie charts for each mouse strain. Two SNPs per allele were measured and showed identical allele frequencies. B6 SNP bolded, *\**p* < 0.01, one-way ANOVA compared to WT B6/BALB mice.

expressed genes (DEGs) in 923[del/del] skins, in contrast to 157 and 222 genes in 923[large/large] and $Ivl^{-/-}$ skins, respectively (FDR < 0.05, log2(FC) ≥ |2|; FC, fold change) (Fig. 3c and Supplementary Fig. 5a). Strikingly, 5 of the 6 DEGs in 923[del/del] (*Rps3a3*, *Hist1h2aI*, *Gm16011*, *Rsp3a1*, and *Ivl*) were also differentially expressed in the same directions in 923[large/large] skins (Fig. 3c and Supplementary Fig. 5b). *Gm15264* was only differentially expressed in 923[del/del] mice. As *Rps3a1*, *Rps3a3*, *Hist1h2aI*, and *Gm16011* were also differentially expressed in the same directions in $Ivl^{-/-}$ skins, their observed differential expression in the 923 deletion lines are likely secondary effects associated with decreased *Ivl* expression in the two enhancer deletion skins. Together, our in vivo results identify a functional role for the 923 enhancer in the regulation of *Ivl* target gene expression.

The effect of the enhancer deletion on the expression of its most proximal gene, *Ivl*, motivated us to further examine local transcriptional effects with respect to only the EDC. RNA-seq analyses of the EDC loci identified one DEG (*Ivl*) in 923[del/del], 10 in 923[large/large], and 10 in $Ivl^{-/-}$ skin (Supplementary Fig. 6) (FDR < 0.05, log2(FC) ≥ |2|). *Smcp*, the most proximal gene 3′ of 923 (Fig. 3b), was below our detection limits for this assay, consistent with the low numbers of transcripts detected in whole-skin scRNA-seq[27]. We validated the significance of decreased *Ivl* expression in all 3 mouse lines compared to WT via qPCR (ANOVA; Tukey post hoc, $p < 0.05$) (Fig. 3d). By contrast, 923[large/large] mouse skins also exhibited significantly decreased expression of *Lce6a*, the most proximal gene 3′ of the deletion, which was not observed in the other 2 mouse lines. This suggests the presence of an additional as yet uncharacterized enhancer region for *Lce6a* that was also deleted in the 923[large] allele.

**Deletion of 923 enhancer affects the chromatin landscape in the EDC.** We next sought to determine the effect of the 923 enhancer deletion on the EDC chromatin landscape using ATAC-seq (assay for transposase accessible chromatin using sequencing)[28]. Six differentially accessible regions (DARs) were found in both 923[del/del] and 923[large/large] keratinocytes compared to WT. Interestingly, three of these shared DARs were within or near (<250 kb) the EDC with two less-accessible DARs and one more-accessible DAR found in both 923[del/del] and 923[large/large] keratinocytes (Supplementary Fig. 7 and Supplementary Tables 5 and 6) (FDR < 0.05, log2(FC) ≥ |2|). Although none of the shared DARs correspond to differential changes in gene expression as determined by RNA-seq, our findings demonstrate an effect for the loss of the 923 enhancer on opening local EDC chromatin accessibility in newborn keratinocytes. Together, both enhancer deletion lines demonstrate a requirement for the enhancer to positively regulate *Ivl*, and additionally for the 923[large] allele for *Lce6a* expression, and facilitate local chromatin accessibility.

**923 enhancer regulates *Ivl* target gene expression in an allele-specific (*cis*) manner.** We next determined if the 923 enhancer regulates gene expression in *cis*. To do this, we performed allele-specific gene expression assays for *Ivl* and *Lce6a* in hybrid mouse skin. Allele-specific *Ivl* and *Lce6a* transcripts were distinguished using two informative SNPs for either B6 or BALB/cBYJ (or BALB) allele in hybrid B6;BALB mouse tissue. Targeted sequencing of *Ivl* cDNA from 923[del]B6/BALB and 923[large]B6/BALB hybrid mouse skin revealed a significantly lower proportion of B6 transcripts from the 923[del] and 923[large] alleles (13%) compared to the proportion of B6 transcripts from the WT allele observed in hybrid control B6/BALB mice (50%) (Fig. 3e) (ANOVA, Tukey post hoc, $p < 0.01$). Moreover, a significantly lower proportion of *Lce6a* B6 transcripts (3%) was observed in 923[large]B6/BALB hybrid skin compared to 42% and 40% observed

in the hybrid control B6/BALB and the 923[del]B6/BALB skin, respectively (ANOVA, Tukey post hoc, $p < 0.01$). This further supports the hypothesis for the loss of an additional regulatory enhancer in the 923[large] allele to regulate allele-specific *Lce6a* expression. Together, our genetic findings identify *cis* regulation by the 923 enhancer for *Ivl*, thus establishing a 923 enhancer:*Ivl* regulatory module for the epidermis.

**The human 923 sequence enhances expression from the *IVL* promoter with both elements exhibiting population-specific regulatory activities.** We next translated this new functional knowledge of the 923 enhancer:*Ivl* regulatory module found in mice to determine the genomic sequences and variants that drive increased *IVL* expression in the positively selected CEU-*IVL* human haplotype. Enhancer rs4845327, promoter rs1854779, and intronic rs7539232 and rs11205132 were identified as positively selected signals by iSAFE and are *IVL* eQTL alleles specific to CEU-*IVL* (Fig. 4a and Supplementary Table 1). We hypothesized increased regulatory activity within the promoter and enhancer for CEU-*IVL* in comparison to common JPT/CHB and YRI haplotypes. We performed luciferase assays to assess regulatory activities for population-specific *IVL* promoters with and without the enhancer for CEU-, JPT/CHB-, and YRI-*IVL*. Clones for *IVL* promoter alleles included the first noncoding exon and intron of known collective regulatory activity[29]. The CEU-*IVL* promoter/intron allele exhibited significantly higher luciferase activity than the JPT/CHB and YRI alleles in both proliferating and differentiated keratinocyte cell culturing conditions (Proliferating: CEU vs. JPT/CHB $p = 0.003$, CEU vs. YRI $p = 0.025$; Differentiated: CEU vs. JPT/CHB $p = 0.008$, CEU vs. YRI $p = 0.001$), consistent with the GTEx annotated SNPs for increased *IVL* expression (Figs. 4b and 2c). The addition of the respective population-specific 923 enhancer resulted in a further increase in luciferase reporter activities for all tested alleles compared to the promoter/intron only, and higher in differentiated cells where *IVL* is endogenously expressed in skin tissue (Fig. 4b). However, decreased luciferase activity for the cloned JPT/CHB enhancer/promoter/intron allele was observed in comparisons to CEU and YRI alleles (Proliferating: CEU-JPT/CHB, $p = 0.003$; YRI-JPT/CHB, $p = 0.024$; Differentiated: CEU-JPT/CHB, $p = 0.047$; YRI-JPT/CHB, $p = 0.051$) (Fig. 4b). The JPT/CHB allele is associated with relatively decreased *IVL* expression as shown in GTEx (Supplementary Fig. 8 and Supplementary Table 7). Together, our results identify increased regulatory activity for the *IVL* promoter/intron allele with an additive effect by the enhancer for the positively selected CEU-*IVL* haplotype.

We next sought to determine differential transcription factor binding for the reference and alternate SNP alleles that underlie the observed regulatory activities for the population-specific *IVL* alleles (Supplementary Table 8). Using multiple transcription factor motif analyses, we identified four SNPs that were predicted to impact differential binding by keratinocyte-specific transcription factors. The SNPs included the two positively selected iSAFE SNPs, *IVL* enhancer (rs4845327) and the *IVL* promoter (rs1854779), as well as two additional SNPs in the *IVL* enhancer (rs1974141) and first intron (rs7517189) (Fig. 4c). ZNF263 is predicted to bind to CEU 923 enhancer rs1974141-A in contrast to MAZ binding to the alternate rs1974141-G (JPT/CHB), suggesting either higher activation by ZNF263 or loss of a MAZ-mediated repressive effect for the relative increased *IVL* expression and enhancer activity observed in the CEU cloned allele. IRF1 is predicted to bind to the positively selected enhancer SNP, rs4845327-G, and negatively affect SOX10 binding for the alternate T allele, whereas NFIC is predicted to bind to the positively selected promoter SNP, rs1854779-T, in contrast to

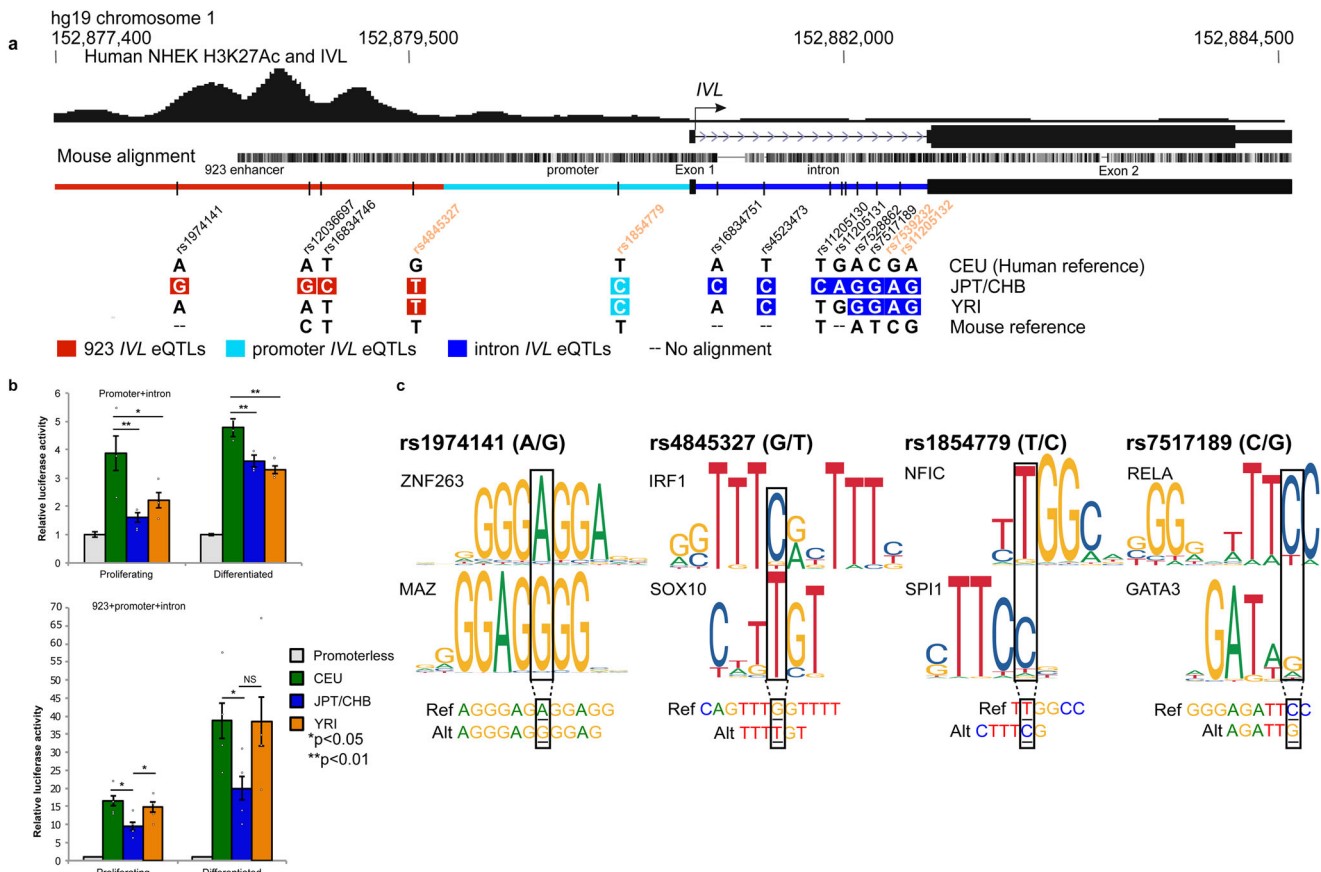

**Fig. 4 Population-specific 923 enhancer: *IVL* promoter/intron alleles and eQTLs impact regulatory activity. a** Schematic of population-specific human 923 enhancer (red), *IVL* promoter (light blue), 1st exon, and intron (dark blue) alleles with *IVL* eQTLs. Positions of variants not to scale. Colored boxed variants are associated with relatively lower *IVL* expression levels to human reference. Positively selected iSAFE SNPs for CEU in orange. **b** Luciferase assays of population-specific alleles for the *IVL* promoter/intron + respective enhancer reveals that the CEU promoter/intron allele has the highest regulatory activity (Proliferating: $p = 0.003$ vs. JPT/CHB; $p = 0.025$ vs. YRI; Differentiated: $p = 0.008$ vs. JPT/CHB; $p = 0.001$ vs. YRI). The addition of the enhancer confers higher reporter expression with more activity in CEU ($p = 0.003$) and YRI ($p = 0.024$) than JPT/CHB and especially in differentiated cells for CEU vs. JPT/CHB ($p = 0.047$). Mean ± SEM of n independent experiments ($n = 3$, *IVL* promoter/intron) and ($n = 4$, 923 enhancer + *IVL* promoter/intron), one-way ANOVA followed by Tukey's HSD. **c** Position weight matrices (PWM) for candidate transcription factor binding motifs in reference and alternate alleles at rs1974141, rs4845327, rs1854779, & rs7517189 associated with population-specific regulatory activity. Strand sequence with SNP included for comparison to PWM.

SPI1 for the alternate rs1854779-C allele. RELA is predicted to bind to intronic rs7517189-C compared to GATA3 for the G allele. Using publicly available ENCODE ChIP-seq datasets, we demonstrated preferential binding in vivo for ZNF263 (A), MAZ (G), NFIC (T), and SPI1 (C) to the predicted alleles with polymorphic sites (Supplementary Fig. 9 and Supplementary Table 9). These findings for allele-specific, transcription factor binding at several loci in different cell types are consistent with the following allele-specific findings that we observe for the positively selected CEU-*IVL* haplotype: (1) increased *IVL* expression, (2) increased regulatory activities for the enhancer and promoter, and (3) preferential ZNF263 and IRF1 predicted binding to rs1974141-A and rs4845327-G in the enhancer and NFIC binding to rs1854779-T in the promoter. In summary, we identify positive selection for a *cis*-regulatory module for increased *IVL* that underwent a selective sweep out-of-Africa and highlights human skin barrier adaptation.

## Discussion

Our work thus far has identified recent skin barrier evolution in modern human populations, having discovered several population-specific signals of positive selection within the EDC

and selection for two neighboring haplotypes, *LCE1A-SMCP* and *IVL* in CEU. The association of the CEU-*IVL* haplotype with relatively increased *IVL* gene expression and the overlap to the known epidermis-specific enhancer 923 led us to examine a functional role for the 923 enhancer to regulate *Ivl* expression in vivo. Deletion of the orthologous mouse 923 enhancer via CRISPR/Cas9 genome editing identified *Ivl* as the primary target gene of 923 with regulation in *cis*, thus establishing a 923 enhancer-*Ivl* gene regulatory module. The functional significance of this genetic module in humans was further revealed with our discoveries of population-specific 923 enhancers that "boosted" regulatory activity. Together, we establish the significance of a paradigmatic regulatory module for a *cis*-regulatory enhancer/ target gene in human evolution.

Our understanding of the molecular mechanisms that drive enhancer activity[30,31] and the comprehensive genomic elucidation of putative enhancers for a vast array of tissue types, as well as single cells, has exponentially grown since the inception of the "enhancer" concept in 1981[32]. Yet, we are challenged to embark on more rigorous investigations to functionally validate enhancers and correctly assign their predicted target genes[31]. Even more challenging is the growing body of recent reports demonstrating a lack of phenotypes for enhancer deletions, even for

putative developmental enhancers, suggesting enhancer redundancy consistent with the ENCODE findings of 2–3 enhancers per target gene[33–35]. As *Ivl* expression was not completely lost in our enhancer knockout mice, there is likely redundancy in regulation of *Ivl* yet we have firmly established the 923 enhancer as a prominent proximal enhancer. In keratinocytes, 5C studies identified physical interactions between the *Ivl* promoter and enhancers located in the first topologically associated domain (TAD) of the EDC, presenting additional putative regulators[36]. Our discovery for the 923 enhancer to regulate *Ivl* expression was premised on extensive previous analyses of the 923 enhancer, where we identified its sequence conservation across mammalian phylogeny, DNaseI hypersensitivity in primary keratinocytes, regulatory activity observed in reporter assays[7], tissue-specific (epidermis) activity using transgenic mice, and physical chromatin looping contacts determined by chromatin conformation capture studies[25]. Such studies provide a framework to prioritize such ongoing and future enhancer in vivo deletion studies. Furthermore, our CRISPR/Cas9 methodology led to the serendipitous generation of a larger deletion (923^large), which displayed decreased *Lce6a* expression that generates the hypothesis for additional enhancers that were deleted in the 923^large allele.

Interestingly, the backbone of the CEU-*IVL* haplotype appears to have emerged in Africa and rose to near-fixation frequency in CEU and other European populations. This selective sweep for CEU-*IVL* is associated with increased *IVL* expression, which we tied to the promoter/intron variants with an additive effect upon the inclusion of the enhancer allele. The effects were further predicted to impact preferential transcription factor binding for ZNF263, IRF1, and NFIC in contrast to MAZ, S0X10, and SPI1 binding in the alternative alleles for rs1974141 and rs4845327 enhancer, and rs1854779 promoter SNPs, and for which preferential binding for these TFs (transcription factors) at single polymorphic sites have been observed in ENCODE ChIP-seq data. Future experiments are needed to isolate and further determine the functional effects of each SNP on *IVL* expression and even on other cell types. In fact, rs1854779 was recently determined to be associated with white blood cell count and was discovered by including diverse human populations to comprehensively determine phenotypic traits[37]. This, together with our human evolutionary findings, generates a hypothesis for a human skin diversity SNP to rapidly evolve and modulate the functional interface of the skin and immune systems.

Our discovery of enhancer variation for human skin barrier function further contributes to a growing body of research that highlights selective events in human history targeted at enhancers, such as lactase persistence[38–42] and immune function[43]. Furthermore, our finding for enhancer evolution further supports genetic innovation for human skin evolution that until now has been reported for only a few genes in the EDC, including *IVL*[44]. *IVL* has undergone extensive evolution in primates, with recent and continuing expansion of the number of repeats in human *IVL*[45–52]. However, targeted deletion of the *Ivl* gene in mice revealed no overt phenotype in barrier-housed conditions[26], similar to our 923^del/del and 923^large/large homozygous mice (Supplementary Fig. 4). Together, this supports a tolerance for decreased, and even lack of, Ivl in barrier-housed conditions, consistent with human population-specific selection for differential *IVL* expression that warrants further investigation. The findings for a selective sweep for increased *IVL* observed in European populations that live in Northern latitudes suggests a possible link between cutaneous vitamin D production and *IVL* expression as a mechanism of environmental adaptation for the skin barrier. Vitamin D is known to stimulate the differentiation of keratinocytes[53–55] and promote the expression of *IVL*[56,57]; it is possible that high rates of vitamin D deficiency in modern populations[58,59] could contribute to reduced skin barrier integrity and, subsequently, disease states including asthma[60].

Selective sweeps for haplotypes, CEU-*LCE1A-SMCP* with increased *HRNR* and CEU-*IVL* with increased *IVL* expressions, suggest a benefit for higher protein dosage for the evolving skin barrier. This notion is supported by (1) the absence of common deleterious (albeit truncating) variants in strong linkage within these haplotypes and (2) the highly repetitive nature of HRNR and IVL that lengthens the protein structures observed across mammalian and primate phylogenies[45–52,61] and anticipated trajectories in humans. The functional benefits for increased HRNR and IVL have yet to be functionally determined, yet it can be speculated that protein dosage modulation provides an innovative strategy to calibrate skin barrier function, i.e., permeability, to the environment. Indeed, weakened epidermal barrier function is a hallmark of atopic dermatitis, a common inflammatory disease, owing to the discovery for >100 loss-of-function variants for the highly repetitive *FLG* gene that are population-specific, and an increase in prevalence worldwide[62].

In summary, our results highlight the significance of genetic variation attributed to differential gene expression in a population-specific manner and heighten our awareness for human population-specific evolution of the epidermis. Furthermore, our work provides a framework with which we can examine genetic variation in enhancer:target gene modules for recent adaptive traits.

## Methods

**CMS and iSAFE analyses.** CMS scores for all SNPs in the EDC (hg18; chr1:150,198,268-151,892,013) in CEU, YRI, and JPT/CHB populations ($n = 90$ individuals per population) were downloaded (https://www.broadinstitute.org/cms/results; download date August 17, 2015). *p*-values were calculated for the genome-normalized CMS scores using pnorm in R. Benjamini-Hochberg FDR was calculated for each population using the p.adjust function, method = "BH" in the fdrtool package in R. iSAFE (https://github.com/alek0991/iSAFE)[21] was used to identify positive selection in 1KGP Phase 3 phased data (http://ftp.1000genomes.ebi.ac.uk/vol1/ftp/release/20130502) (hg19; chr1:151,896,000-153,612,000). Sample cases were defined as all subjects belonging to the population of interest (CEU, FIN, IBS, YRI or JPT, and CHB) and sample controls defined as subjects belonging to all other superpopulations. All other arguments were run using default settings.

**Haplotype block construction and LD analysis.** Haplotypes were identified using LDLink (https://ldlink.nci.nih.gov) by querying the LDproxy module for the SNP with the highest CMS score in the relevant population(s). Haplotypes were defined as the set of proxy variants with $r^2 \geq 0.8$. Pairwise and small-scale analyses of LD were performed using the LDpair module, again defining LD as $R^2 \geq 0.8$.

**Global allele frequency and eQTL analysis.** Allele frequencies of EDC SNPs from the twenty-six 1KGP populations were obtained from Ensembl v. 81[63,64]. Geographical latitude for each population was determined by the latitude of the city from which 1KGP, ACPOP (Northern Sweden), Genome of the Netherlands release 5, Genetic variation in the Estonian population, and the Danish reference pan genome population data for tagging SNP rs4845327 were collected. The relationship between latitudes and allele frequencies was analyzed using linear regression using the lm function in R. The direction and strength of the correlation between allele frequency and latitude was determined using Pearson's correlation coefficient ($\rho$) using the cor function in R. Geographic plots for the global distribution of allele frequency was generated using ggplot2 in R. SNPs were queried as eQTLs in sun-exposed or not sun-exposed skins using GTEx portal (http://www.gtexportal.org; V8 release).

**Generation of 923 enhancer knockout alleles in mice by CRISPR/Cas9 genome editing.** Deletion of the 923 enhancer was targeted using two small guide RNAs to facilitate Cas9-mediated double-stranded breaks on either side of the orthologous mouse 923 enhancer sequence. *LoxP* insertions at these flanking sgRNA-targeted sites were generated using two single-stranded oligodeoxynucleotides (ssODNs), containing *loxP* sites with specific 80 bp homology arms on either side of *loxP* and restriction enzyme sites (SphI for 5′ end, HindIII at 3′ end), which were simultaneously introduced via site-directed homologous recombination (Integrated DNA Technologies, Coralville, IA). Sequences of sgRNAs, ssODNs, and mouse alleles are listed in Supplementary Information (Supplementary Table 10). Three rounds of injection in 779 zygotes were performed having confirmed target specificity using in vitro pilot studies prior to zygote injection. Founders were

initially screened for large deletions via PCR using flanking primers designed outside the homology arms and 5′ and 3′ loxP-specific primers that were resolved on 1% agarose gel electrophoresis. Of the 779 C57BL6/6XCBA hybrid zygotes injected, 80 $F_0$ newborns were recovered, of which 75 survived to weaning age. Seven out of 80 mice whose 923 allele size deviated from the WT allele were identified via PCR (8.75% targeting efficiency) with further analysis by Sanger sequencing. Two 923 enhancer knockout alleles were confirmed. The 923[del] allele contains an ~1250 bp deletion of the 923 enhancer and the 5′ loxP site. The 923[large] allele includes the 923 deletion flanked by both 5′ and 3′ loxP sites and a 40 kb deletion between proximal genes Lce6a and 923, including the genes Smcp and 2210017I01Rik. All mice were group-housed in cages with bedding and nesting material in pathogen-free, barrier facilities (65–75 °C and 40–60% humidity) with a 12 h light/dark cycle at Washington University School of Medicine (St. Louis, MO). All animal procedures were approved by the Division of Comparative Medicine Animal Studies Committee at Washington University in St. Louis School of Medicine. All animal work was conducted in accordance with the Guide for the Care and Use of Laboratory Animals of the National Institutes of Health. Morning observation of a vaginal plug was designated as embryonic day (E) 0.5. Both 923[del] and 923[large] mouse lines were backcrossed at least 7 times to the C57BL/6 background to generate isogenic strains and to exclude potential off-target effects arising from CRISPR/Cas9 editing. Measurements for each of the molecular assays for the mice were taken from distinct samples. A list of all primers used throughout this study are provided in Supplementary Information (Supplementary Table 11).

**Histology.** Dorsal skin was excised from 8-week-old mice and preserved in 4% paraformaldehyde (Electron Microscopy Sciences, Hatfield, PA) prior to paraffin sectioning. Sections were stained with hematoxylin and eosin by the Washington University Developmental Biology Histology Core. Slides were imaged on a Nikon Eclipse 80i brightfield microscope (Nikon, Tokyo, Japan).

**Dye penetration barrier assays.** Barrier assays were performed with X-gal solution incubations for at least 4 h at 37 °C[65]. Images were captured on a CanoScan 5600 F scanner (Canon, Melville, NY).

**Cornified envelope preparations.** Epidermis (cut into 1 cm$^2$ pieces) was incubated at 95 °C in a solution of 2% SDS to obtain cornified envelopes in a single-cell suspension. The suspension was placed on a slide and inspected using phase contrast light microscopy.

**Trans-epidermal water loss assay.** TEWL was measured on the abdominal skin surface of newborn mice using the nail attachment of a VapoMeter (Delfin Technologies, Kuopio, Finland, courtesy of Jeff Miner).

**Immunofluorescence.** Fresh sections were fixed in 4% paraformaldehyde (Electron Microscopy Sciences, Hatfield, PA) prior to permeabilization with 0.1% Triton X-100 and subsequent antibody incubation. The following antibodies were used for immunofluorescence: rabbit IVL (4b-KSCN, 1:200) and chicken K14 (5560, 1:500) custom antibodies (courtesy of J. Segre), goat anti-rabbit (Alexa Fluor 488 #A-11008, 1:500), and goat anti-chicken (Alexa Fluor 594 #A-11042, 1:500) IgG antibodies (Life Technologies, Frederick, MD), and DAPI counterstained (Slow-Fade Gold antifade reagent) (Life Technologies). Fluorescent microscopy was performed on Zeiss AxioImager Z1 and captured with AxioCam MRc and Axio-vision software (Carl Zeiss, Stockholm, Sweden), or imaged on a DMI3000 B (Leica, Wetzlar, Germany) and captured with the QIClick camera (QImaging, Surrey, BC, Canada).

**RNA-seq.** Total RNA from whole skin was isolated by TriZol extraction (Life Technologies, Frederick, MD). Ribo-zero (ribosome-depleted) RNA sequencing libraries were prepped according to the manufacturer's library kit protocol, indexed, pooled, and sequenced on Illumina HiSeq 3000 (1X 50 bp) by the Washington University Genome Technology Access Center.

Basecalls and demultiplexing were performed with Illumina's bcl2fastq software v2.20 with a maximum of one mismatch in the indexing read. RNA-seq reads were then aligned to the Ensembl release 96 top-level assembly with STAR version 2.0.4b. Gene counts were derived from the number of uniquely aligned unambiguous reads by Subread:featureCounts version 1.4.5. Sequencing performance was assessed for the total number of aligned reads, total number of uniquely aligned reads, and features detected. The ribosomal fraction, known junction saturation, and read distribution over known gene models were quantified with RSeQC version 2.3. All gene counts were then imported into the R/Bioconductor package EdgeR version 3.22.0, and TMM normalization size factors were calculated to adjust for samples for differences in library size. Ribosomal genes and genes not expressed in the smallest group size minus one samples greater than one count-per-million were excluded from further analysis. The TMM size factors and the matrix of counts were then imported into the R/Bioconductor package Limma version 3.36.5. Performance of the samples was assessed with Spearman correlations, a multi-dimensional scaling plot, and hierarchical clustering. Weighted likelihoods based on the observed mean–variance relationship

of every gene and sample were then calculated for all samples with the voomWithQualityWeights. The performance of all genes was assessed with plots of the residual standard deviation of every gene to their average log-count with a robustly fitted trend line of the residuals. Differential expression analysis was then performed to analyze for differences between conditions and the results were filtered for only those with Benjamini-Hochberg false-discovery rate adjusted $p$-values ≤ 0.05, and a log2(FC) $\geq |2|$.

The R/Bioconductor package heatmap3 version1.1.6 and Pathview version 1.18.2 was used to display heatmaps or annotated KEGG graphs across groups of samples for each GO term or KEGG pathway, respectively, with a Benjamini-Hochberg false-discovery rate adjusted $p$-value less than or equal to 0.05.

Real-time qPCR on cDNA (generated using SuperScript II reverse transcriptase (Thermo Fisher Scientific) using TaqMan Gene Expression Assay was performed in triplicate (Applied Biosystems, Life Technologies, Carlsbad, CA) and normalized to β2-microglobulin. Only CT values with single peaks on melt-curve analyses were included.

**ATAC-Seq.** ATAC-seq was performed on 75,000 epidermal cells from each mouse with variations[28]. Cells were lysed for 5 min in 37.5 μl ice-cold buffer (10 mM TrisHCl, 10 mM NaCl, 2 mM MgCl$_2$, 0.5% IGEPAL CA-630). Cells were then pelleted at 500g (4 °C) for 15 min. Lysis buffer was replaced with transposition reaction mix (12.5 μl TD (2X reaction buffer from Nextera Kit, Illumina, San Diego California, USA), 2.5 μl TDE1 (Nextera Tn5 Transposase from Nextera kit, Illumina), and 10 μl H$_2$O) and samples were incubated for 1 h at 37 °C. Samples were purified using Qiagen MinElute PCR Purification Kit (Qiagen, Valencia, CA) and PCR-amplified. Adapter dimer bands were removed using AMPure XP bead treatment (Beckman Coulter, Brea, CA). All Samples exhibited the expected nucleosome periodicity as assayed by High Sensitivity ScreenTape (Agilent Technologies, Santa Clara, CA). Samples were sequenced via Illumina HiSeq2500 (2X 50 bp). An average of 93.4% of the reads were mapped with 14.3–39.8 million qualified reads per sample with only an average of 6.8% mitochondrial reads (Table S18). Prior to sequencing, all samples exhibited the expected periodicity of insert length and were enriched for reads at transcription start sites. ATAC-seq data were processed using the ENCODE ATAC-seq processing pipeline, using Caper with Conda (https://github.com/ENCODE-DCC/atac-seq-pipeline)[66]. Reads were mapped using Bowtie2 version 2.3.5.1, and filtered to remove unmapped reads, duplicates, and reads mapping to chrM. Peaks were called on each replicate using MACS2[67]. Biological replicates were included if both the rescue and self-consistency IDR values per genotype were below (or very near to) 2 (Supplementary Table 12). Differential accessibility was assessed using EdgeR[68] within the DiffBind R package version 2.12.0 (http://bioconductor.org/packages/DiffBind/) (FDR < 0.5, log2(FC) $\geq |2|$).

**Allele-specific gene expression.** RNA from newborn whole skins was isolated as described above from C57Bl6 and BALB/cBYJ wild-type mice and from [C57Bl6]/[BALB/cBYJ], [923[del]/[BALB/cBYJ], and [923[large]/[BALB/cBYJ] hybrid mice. RNA was DNaseI treated and reverse transcribed into cDNA using Invitrogen SuperscriptII Reverse Transcriptase (Invitrogen, Carlsbad, CA); 260 bp amplicons targeting the Ivl, 2210017I01Rik, and Lce6a genes were amplified from cDNA using NEBPhusion High Fidelity PCR 2X master mix (New England BioLabs, Ipswich, MA) (Amplicon Sequences in Supplementary Table 11). PCR products were purified on Qiagen QIAquick PCR columns (Hilden, Germany), A-tailed with 1 mM dATP and NEB Taq polymerase for 20 min at 72 °C (New England BioLabs, Ipswich, MA) followed by an additional column purification on Qiagen MinElute columns. Next-generation sequencing-compatible adapters were ligated to A-tailed PCR products in molar excess using the LigaFast DNA ligase kit (Promega, Madison, WI) at room temperature for 20 min. Products were size-selected using Agencourt AMPure XP beads (Beckman Coulter, Brea, CA) at 1.2X product volume to remove excess adapter. Adapter-ligated products were quantitated using Qubit dsDNA High Sensitivity Assay kit (Life Technologies, Carlsbad, CA). To determine appropriate PCR amplification cycle number to avoid over amplification, quantitative PCR was performed using 2X NEB Phusion High Fidelity PCR master mix, 0.5 μM PCR primer1, 0.01 μM PCR primer2, 0.5 μM unique Index Primer, 100X SYBR Green and 50X ROX Dye, and 2 ng DNA in 10 μl. PCR cycle number for each template was determined by identifying the cycle where ¼ max fluorescence was reached; 10 ng of each library was amplified using the same reaction, without SYBR and ROX, scaled up to 50 μl. Amplified libraries were size-selected again to remove any remaining adapter dimer using 1X concentration of AmPure beads. All libraries were pooled at equal molar ratio and sequenced as a spike-in to a 2 × 150 bp MiSeq sequencing run, averaging 154,000 reads per sample. Demultiplexed reads were mapped using Bowtie2 version 2.3.5.1 and visualized using the IGV viewer. The proportion of nucleotides at each informative SNP in the amplicon was calculated by IGV. Primers are listed in Supplementary Information (Supplementary Table 11).

**Luciferase assays.** Population-specific alleles for the IVL promoter, noncoding first exon, and intron were cloned from human gDNA by PCR using primers 5′-GGATCCGATAGGTTCTAGGGGTATAGTGG/5′-AAGCTTCTTAGAAGCTACTGTCAACCTG. PCR products were digested with BamHI and HindIII and

cloned into the BglII/HindIII site in pGL3 (Promega) to yield pGL3-IVLpromoter and confirmed by Sanger sequencing. The Gateway cassette B was then cloned into the SmaI site to yield pGL3-IVLpromoter-GW. The 923 enhancer region was amplified from human gDNA using primers 5′- GGGGACCACTTTGTACAAG AAAGCTGGGTGAAGAACAGTGAATTTTACGACC/5′- GGGGACAAGTT TGTACAAAAAAGCAGGCTAGACATTCTGCTGCTGGACA and introduced into pDONR221 by BP recombination using BP Clonase II (Invitrogen, Thermo Fisher Scientific). Haplotype-specific variants were confirmed by Sanger sequencing. Enhancer alleles were then introduced into pGL3-*IVL*promoter-GW by LR recombination using LR Clonase II (Invitrogen, Thermo Fisher Scientific) to yield pGL3-*IVL*promoter-923. Enhancer alleles were then introduced to pGL3 by LR recombination. Luciferase reporter assays were performed in the mouse SP-1 keratinocyte cell line with measurements performed at 48 and 72 h post-differentiation for proliferating and differentiated cells, respectively, using a Glo-max luminometer (Promega)[7,25]. Significance was determined using one-way ANOVA followed by Tukey's HSD.

**Transcription factor binding predictions**. Transcription factor binding prediction was determined using transcription factor motifs considering the position and weight of the nucleotide for a given binding motif and evidence of transcription factor expression in keratinocytes (Human Protein Atlas)[69,70] and ENCODE ChIP-seq binding (where possible). Both reference and alternate nucleotides for a given SNP were rigorously queried and centered at position 25 in a 50 bp window were analyzed with PROMO 3.0[71,72] and ConSite (http://consite.genereg.net/cgi-bin/consite). PROMO 3.0 with TRANSFAC version 8.3 considered only human sites and human factors with 15% maximum matrix dissimilarity rate and ConSite utilized the option for all transcription factor profiles with a minimum specificity of 10 bits and transcription factor score cutoff of 80% in a single sequence. JASPAR[73] with a relative profile score threshold of 80% was used to analyze each SNP allele (reference and alternate independently) centered at position 15 in a 30 bp window. HAPLOREG v.4[74] was queried for a given SNP rsID. ENCODE ChIP-seq matrix considered *Homo sapiens* transcription factors that were present at the SNP location in skin cell lines, keratinocyte or foreskin keratinocyte primary cell lines, or suprapubic skin tissue[75]. Transcription factors predicted were checked for positive antibody staining in keratinocytes from primary data in the Human Protein Atlas[69,70]. We interpreted transcription factor binding that satisfied JASPAR plus either ConSite, PROMO 3.0, or HAPLOREG predictions.

**Allele-specific ChIP-seq analysis**. All ChIP-Seq data, including IDR thresholded peaks bed and bam alignment files were downloaded from ENCODE (accessions list, Table S19). Variant calls in vcf format were downloaded from ENCODE (HepG2: ENCSR319QHO, K562: ENCSR053AXS), the Platinum Genomes project[76], and the HEK293 genome project (http://hek293genome.org/v2/)[77] and if necessary converted to hg38 using liftOver. Variants were further filtered to only include heterozygous variants. Sequences for each ChIP-seq peak were extracted using bedtools and the hg38 UCSC genome reference. Motifs for each TF from JASPAR 2020[73] (MAZ: MA1522.1, NFIC: MA0161.1, SPI1: MA0080.1, ZNF263: MA0528.2) and used FIMO (part of the MEME suite v5.3.3)[78] were downloaded to search for occurrences of each motif with max-stored-scores set to 1E-8. Since the appropriate *p*-value threshold is dependent on the motif length and information content, *p*-values were scanned for each motif search across the values 1E-5, 5E-4, 1E-4, 5E-3, and 1E-3. For each motif, the *p*-value was set based on whether there was an average of 0.5–1 motifs found per ChIP-seq peak across each of the datasets for a given TF (Supplementary Table 13). Bedtools (v2.26.0, https://bedtools.readthedocs.io/) was used to overlap identified motifs with heterozygous SNPs using the variant calls (.vcf file) from the appropriate cell line. Samtools (v1.3.1, http://www.htslib.org/) view with the -L flag was used next to extract all alignments that overlapped identified ChIP-seq peaks. Aligned reads were overlapped to identify each SNP, manually curated to ensure the SNP corresponded to the queried variant, and counted for the number of reads supporting each base. Further, only SNPs with coverage ≥5 and within 50 bp of the peak center were retained. Results from IRF1 and RELA are not included since no SNPs met these criteria.

**Ethics**. Cloned human alleles was obtained from prior human research[79] according to the Declaration of Helsinki with provided written informed consent. The study was approved by the Washington University in St. Louis Institutional Review Board (#201109075).

**Reporting summary**. Further information on research design is available in the Nature Research Reporting Summary linked to this article.

## Data availability
All data supporting the findings of this study are available within this article and the Supplementary Information. Raw RNA and ATAC sequencing data are available in NCBI Gene Expression Omnibus (GEO) using the accession code GSE158870. The following ENCODE ChIP datasets were used: GM12878-MAZ (ENCSR903MVU, ENCSR000DZA), HEK293-MAZ (ENCSR290SSQ), HepG2-MAZ (ENCSR700PNE,

ENCSR000EDN), K562-MAZ (ENCSR163IUV, ENCSR643JRH), HEK293-ZNF263 (ENCSR000EVD), HepG2-ZNF263 (ENCSR313MMD), K562-ZNF263 (ENCSR000EWN), GM12878-NFIC (ENCSR000BRN), K562-NFIC (ENCSR796ITY), GM12878-SPI1 (ENCSR000BGQ), and K562-SPI1 (ENCSR000BGW). Links to the ENCODE datasets and the reporting summary are available in the Supplementary Information. Source data are provided with this paper.

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

## Acknowledgements

We thank Renate Lewis at the Hope Center for the design and targeting of the gRNAs, Mia Wallace at the Mouse Core facility for the mice, Toni Sinnwell at the Washington University Genome Technology Access Center for the preparations of the RNA-seq libraries and Illumina sequencing, and Justin Fay and Don Conrad for helpful early discussions. The work cited in this publication was performed in a facility partially supported by NCI Cancer Center Support Grant P30CA91842 to the Siteman Cancer Center and by ICTS/CTSA UL1TR002345 and C06RR015502 from National Center for Research Resources (NCRR), a component of the National Institutes of Health (NIH), NIH Roadmap for Medical Research, Society of Investigative Dermatology/Sun Pharma Research Innovation Fellowship (E.A.B., M.E.M.), NHGRI T32HG000045 (Z.A.G., A.D. S. [2020–to date]), T32GM007067 (M.E.M.), R25GM103757 (A.D.S. [2017–2019]), and by NIAMS R01AR065523 and R56075427 (C.dG.S.). This content is solely the responsibility of the authors and does not necessarily represent the official views of NCRR or NIH.

## Author contributions

M.E.M. performed the mouse experiments, data collection, and allele-specific RNA-seq and ATAC-seq analyses. E.A.B. performed the iSAFE and haplotype analyses and human luciferase assays. M.E.M. and E.A.B. prepared the figures and wrote the manuscript. A.D. S. identified human population data sources and provided help with haplotype and transcription factor binding analyses and data visualization. Z.A.G. performed the CMS and statistical analyses. A.M.Q. performed the initial screening of the CRISPR/Cas9

genome editing in cell lines and I.Y.O. completed the screening and initial assays in the mouse including developmental barrier and cornified envelope assays. E.T. and S.A.M. analyzed the RNA-seq data. N.R. validated iSAFE in other human populations, E.G.Y. provided human samples for population-specific alleles for luciferase assays, and J.R.E. and M.E.M. performed the allele-specific ENCODE ChIP-seq analyses. All authors provided critical feedback on the manuscript. C.dG.S. conceived the idea, supervised the experiments, secured funding for the project, and co-wrote and edited the manuscript.

## Competing interests

C.dG.S., M.E.M., E.A.B., Z.A.G., and I.Y.O. are inventors on provisional patent application 63/090,801 submitted by Washington University in St. Louis that covers the compositions and methods for treating skin diseases, disorders, or conditions. C.dG.S. is the founder of Evoly Skin, LLC that is developing new technologies to improve the skin barrier. E.G.Y. is an employee of Mount Sinai and has received research funds (grants paid to the institution) from Abbvie, Almirall, Amgen, AnaptysBio, Asana Biosciences, AstraZeneca, Boerhinger-Ingelhiem, Celgene, Dermavant, DS Biopharma, Eli Lilly, Galderma,Glenmark/Ichnos Sciences, Innovaderm, Janssen, Kiniksa, Kyowa Kirin, Leo Pharma, Novan, Novartis, Pfizer, Ralexar, Regeneron Pharmaceuticals, Inc., Sienna Biopharma, UCB, and Union Therapeutics/Antibiotx; and is a consultant for Abbvie, Aditum Bio, Almirall, Alpine, Amgen, Arena, Asana Biosciences, AstraZeneca, Bluefin Biomedicine, Boerhinger-Ingelhiem, Boston Pharmaceuticals, Botanix, Bristol-Meyers Squibb, Cara Therapeutics, Celgene, Clinical Outcome Solutions, DBV, Dermavant, Dermira, Douglas Pharmaceutical, DS Biopharma, Eli Lilly, EMD Serono, Evelo Bioscience, Evidera, FIDE, Galderma, GSK, Haus Bioceuticals, Ichnos Sciences, Incyte, Kyowa Kirin, Larrk Bio, Leo Pharma, Medicxi, Medscape, Neuralstem, Noble Insights, Novan, Novartis, Okava Pharmaceuticals, Pandion Therapeutics, Pfizer, Principia Biopharma, RAPT Therapeutics, Realm, Regeneron Pharmaceuticals, Inc., Sanofi, SATO Pharmaceutical, Sienna Biopharma, Seanegy Dermatology, Seelos Therapeutics, Serpin Pharma, Siolta Therapeutics, Sonoma Biotherapeutics, Sun Pharma, Target PharmaSolutions, Union Therapeutics, Vanda Pharmaceuticals, Ventyx Biosciences, and Vimalan. Consultant for Abbvie, Aditum Bio, Almirall, Amgen, Asana Biosciences, AstraZeneca, Boerhinger-Ingelhiem, Cara Therapeutics, Celgene, Concert, DBV, Dermira, DS Biopharma, Eli Lilly, EMD Serono, Galderma, Ichnos Sciences, Incyte Kyowa Kirin, Leo Pharma, Pandion Therapeutics, Pfizer, RAPT Therapeutics, Regeneron Pharmaceuticals, Inc., Sanofi, Sienna Biopharma, Target PharmaSolutions, and Union Therapeutics. The other authors declare no competing interests.
