## [Peer Review File · Nature Communications]

Reviewers' Comments:

Reviewer #1:

Remarks to the Author:

Mathyer and co-workers present a complex study of EDC haplotypes, selection, enhancer activity and association with AD. These studies are of interest since they combine genomic and evolutionary approaches to a region of the genome (the EDC) that plays a major role in epidermal differentiation and barrier formation, but which is poorly understood. The likelihood that variants within this region have been under selection in our recent past is not unlikely, but it is nice to see that these are at last being elucidated along with the genes they regulate. There are many interesting findings but as written the manuscript is confusing and needs to be simplified. The studies on AD however, while interesting are severely under-powered and therefore detract from the overall message of the manuscript. Some of the writing in the text and figure legends also needs to be simplified and made clearer.

The authors first determined that two independent IVL haplotypes have been under positive selection in European and Asian populations. These led to an increase in IVL expression that co-evolved with an epidermal-specific enhancer. The ability of this enhancer to regulate IVL, along with the IVL promoter was shown in mice and with luciferase reporter assays. The authors then provided evidence that the IVL enhancer eQTLs associated with decreased IVL and LOF variants in FLG are enriched in atopic dermatitis cases vs. controls.

Other comments:

Since this manuscript will not just be read by evolutionary specialists explain "CMS" and iSAFE and why these particular algorithms were selected for these analyses. Also explain the significance of the scores. P values should be presented for CMS and iSAFE scores, along with adjustment for multiple testing to ensure true significance. Only significant findings should be presented. This could remove some of the multiple hits that are not shared by European and Asian populations. In Table S3 indicate if P values are being presented or something else.

It could be hyperbole to state "As SMCP encodes a sperm mitochondrial cysteine-rich protein, the finding suggests a role for the positively selected SNPs in sperm function" since the variants under selection in that region may be acting on other nearby genes. It is un-necessary for the current study and this sentence should be deleted.

Some of the writing is overly complex: For example Figure 2 legend: "share commonality for both" – wouldn't be more straightforward to say "both carry"? and later on: "A direct and positive correlation between the frequency of rs4845327-G, a tagging SNP for the CEU-IVL haplotype, and latitude reveals a selective advantage for CEU-IVL with increasingly Northern latitudes" – delete "and latitude" since this is what the authors then say is correlated with the tagging SNP.

Page 6. "The CEU-IVL haplotype contains expression quantitative trait loci (eQTLs)" is not entirely correct. It contains alleles of eQTLs. Similarly lower down (eQTLs associated with decreased expression..." and also on page 11.

Page 6. "is located 923 kb from the most 5' EDC gene.." Surely EDC genes are not all in the same 5' to 3' direction – instead refer to proximal or distal chromosomal location.

The change in expression of specific transcripts in the presence of deletion of the enhancer is quite striking, and certainly confirms a role for the enhancer in IVL expression.

Figure 3b shows the location of the orthologous deletion of the enhancer in mouse – however, two different deletions were generated. Show both and explain how these relate to the earlier study of Oh

et al (2014).

Figure 3g. For the FCs for qRT-PCR, state what the mutant mice transcripts were compared with (wt mice?).

The mouse studies are very nice and confirm the role of the 923 enhancer in regulating IVL. Page 8. Presumably the lower level of B6 SNP alleles corresponds to mRNA – however as written this isn't clear.

It isn't clear what the ATAC-Seq studies contribute to the current study, other than showing that the 923 enhancer indeed has this activity, since the regulatory elements do not seem to reside around IVL.

Page 10. When discussing luciferase assays, provide P values obtained when comparing activities of the different enhancers. Describe the SNP alleles that were used in the different constructs. Were any SNP alleles residing the different constructs identified from the selection studies described at the beginning of the manuscript. These data can sometimes be extracted from the manuscript, but they need to be presented in a more palatable form. Describing this clearly would certainly add to the value of the manuscript since these would be candidates within the enhancer that specifically alter IVL expression. Are any transcription factors predicted to map to these SNP alleles?

With the sentence: "Targeted sequencing of *Ivl* and *Lce6a* cDNAs obtained from 923delB6/BALB and 923largeB6/BALB heterozygous mouse skins revealed a significantly lower proportion of B6 SNPs from the 923del and 923large alleles for *Ivl* ..." (again use "skin" and not "skins"). Don't the authors mean B6 SNP alleles rather than SNPs.

The paragraph at the end of page 8 and beginning of page 9 makes no sense – please make clearer, explaining the rationale for the approach.

Page 10: "Our discoveries for the positively selected IVL human haplotypes associated with differential IVL expressions...: change expressions to expression.
Is there conservation between humans and mice at loci where variant alleles affect IVL expression from the 923 enhancer?"

The hypothesis that variants affecting IVL expression are modifiers of LOF FLG mutations is intriguing but confirmation of this will require much larger cohorts.

Table S13 describes variants in the 923 enhancer region found in AD patients. In the column headings of the table, indicate the number of individuals screened in each study. Some of the information on the SNPs needs to be updated. For example the rare allele of rs116579812, described as only being found in AD patients, is actually found in all populations except Europeans at a frequency of about 3-4% (querying dbSNP). This suggests that AD patients were not of European origin entirely. Please provide population breakdown for patients. Moreover, if a variant has an rs number it has already been seen before. Can the authors look for enrichment of rare variants from this enhancer in AD patients versus controls? Getting statistical evidence that rare variants such as these are more common in AD patients than in controls is going to require much larger cohorts and it is suggested that the entire section on AD be deleted if these cannot be queried. For the AD patients studied, is it possible to provide information on which individuals harbored which variant and the haplotypes around IVL that they carried. Where information on IVL levels are available (for example from immunostaining), please indicate as well. AD patients are surely heterozygous carriers for the rare variants – hence the wt allele will impact on IVL expression. This is not discussed.

The request above pertains to this concern as well: The authors note that seven of the 30 variants in the 923 enhancer are IVL eQTLs associated with decreased IVL expression in skin and correspond with two different haplotypes, CEU minor and JPT/CHB. Were these variants found in the same or different

AD patients?

Figure 5 legend: This isn't clear: "e) Decreased IVL protein expression in the AD lesional skins of IVL eQTL and 923 enhancer rs78868757 in AD patient compared to non-IVL eQTL genotyped normal skin". Please rephrase (skins should be skin, what do the authors really mean by: "IVL eQTL and 923 enhancer rs78868757". Perhaps "of" should be replaced by something else.

Similarly, Figure 5 legend f) isn't clear: "f) Venn diagrams highlighting frequencies of IVL eQTLs and FLG LOF variants and was enriched in atopic dermatitis subjects vs. controls, *P<0.05.

Why is "rs78868757" not shown in Figure 5a?

In the discussion the authors mention copy number polymorphism of the IVL gene. Could this account for some of the different levels of expression associated with different haplotypes?

Reviewer #3:

Remarks to the Author:

The manuscript of Mathyer et al reports selective pressure within the epidermal differentiation complex (EDC) locus. SNP data derived from the HapMap consortium from individuals of central European, Han Chinese and Nigerian descent are compared and number of sites of putative positive selection are identified. Involucrin (IVL) was the only gene observed for which positive selection was observed in both Han Chinese and central European individuals. The authors proceed to focus on this locus in subsequent experiments. They propose that this selective effect is mediated through variation in the sequence a previously-described remote regulatory region (enhancer) of the involucrin gene '923'. They also present some evidence to support the hypothesis that variants in this region are modifiers of the penetrance of filaggrin mutations in the aetiology of atopic dermatitis.

This manuscript presents convincing evidence in support of selective pressures affecting the involucrin locus and in support of the '923' element as a regulator of involucrin expression. Whilst this is of interest to those studying this genetic locus it is not particularly novel since there others have previously reported evidence of selection acting upon enhancer elements and the '923' element has previously been reported to be an enhancer.

The population-based data in support of selection affecting the EDC locus is derived from publically available datasets. It was not clear to me how many individuals were included in these datasets, moreover whilst it might be beyond the remit of this study it would be nice to see an independent validation cohort. Comparison with data derived from the 1000 genomes project failed to confirm evidence of positive selection for the Han Chinese population, but did identify a widespread evidence of selection throughout the EDC cluster including the IVL gene in the central European population.

The evidence in support of 923 as a regulatory element is convincing and includes a deletion of the homologous regulatory element in mouse, analysis of allele-specific expression, analysis of the chromatin landscape through ATAC seq and reporter transfection assays. It would be interesting to see an analysis of conserved sequence elements and transcription factor binding sites in this enhancer element, perhaps along with some experiments (e.g. chromatin immunoprecipitation) to establish which transcription factors are binding.

Reviewer #4:

Remarks to the Author:

I like the integration of evolutionary genetics perspective with functional work involving mice models

in a locus that has clear connections to skin disease. It is indeed possible that the authors are on the verge of discovering a new "paradigm" for the evolution of IVL-enhancer variation within the context of its impact on skin disease. However, I think the analysis presented here falls short of explaining the specifics of this paradigm. Please find below some specific comments:

1. The selection tests:

- There are, at this point, dozens of different tests of selection, each with their own sensitivity and accuracy for detecting different types of selective signatures. I think that CMS is a perfectly fine test to use, but it is important to contextualize exactly what it is measuring and what kind of sweep that this test more likely to find (Complete or incomplete, soft or hard, old or recent?)
- It is not clear to me if iSAFE is the right test here. I think this particular algorithm is designed to narrow down specific casual variants, rather than finding new selective signatures. In any case, the reason why these particular tests are used and what is expected to be found should be clarified. This is particularly important given the complex evolutionary history of the EDC.
- I am not sure how the significance of selection is calculated for these tests. It is, I think, important to know whether the values that they are observing for CMS are unusual just for the EDC locus or the whole genome? If they conduct this test, for example, to the LCT locus in CEU or EDAR locus in CHB, what would be the values? In fact, a desirable approach would be to identify the putatively selected haplotypes and using either empirical and/or simulation-based methods to calculate the coalescent, date of the onset of the selection, and the selection coefficient. I know that this may be a little much to ask. However, I think the timing and nature of the adaptation of this locus would be essential to construct the eventual model of this locus' evolution.

2. Enhancer work:

- It is clear both from existing work as well as the authors' mice work that the region of interest is an enhancer. What is more interesting to me that their top SNP actually is hitting this enhancer sequence. This should be stated in the text and Figure 3A.
- I am disappointed that the authors did not reconstruct the allelic variation in the mouse, which may provide a model for evolutionary work as well as AD susceptibility.

3. Modeling the evolution and functional relevance of this locus:

- I am very skeptical about the latitude connection. Any variant that have substantially different allele frequencies in Africa and Eurasia will appear to be correlated with latitude. I think, larger, within-continent datasets is required for a better understanding of this issue.
- I made a quick analysis (in PMID: 31217584 data) and it seems that the putatively selected SNP rs1854779 is very significantly associated with White Blood Cell Counts. How can the authors dismiss immune-mediated phenotypes as a potential adaptive force in shaping the variation in this locus?
- I think there is a lack of specifics when it comes to what exactly the putatively selected haplotype(s) does other than changing expression levels. Moreover, its connection to skin disease has barely been discussed. Even if it is the case that this variation is linked to Vitamin D deficiency, how is lower expression of IVL help with that (weakened Epidermal Layer and more UV penetration?). How is that linked to AD - is weakened epidermal barrier function a hallmark of AD?

4. Some minor suggestions for clarity and organization:

- What is the most 5' EDC gene (line 142)?
- The luciferase assay and the reason as to why these experiments are conducted should be more clearly explained in my opinion, especially within the context of the evolution of the variation in this locus.
- Too much going on in figure 3 and they are not discussed in the main text. The figure can be simplified, or some parts can be put in supplementary

Reviewers' comments:

Reviewer #1 (Remarks to the Author):

1. There are many interesting findings but as written the manuscript is confusing and needs to be simplified. The studies on AD however, while interesting are severely under-powered and therefore detract from the overall message of the manuscript. Some of the writing in the text and figure legends also needs to be simplified and made clearer.

Thank you for the enthusiasm and helpful comments. We simplified the writing of the manuscript text and figure legends. We also addressed the points below. Changes are underlined in the manuscript and marked vertically on the left margin. We removed the studies on AD that are under-powered. In doing so, we emphasize the overall message of our study that reports, for the first time, human skin barrier evolution with the identification of a human haplotype and functional elucidation of a *cis* regulatory module within for increased involucrin that underwent a selective sweep in Northern Europe.

2. Since this manuscript will not just be read by evolutionary specialists explain “CMS” and iSAFE and why these particular algorithms were selected for these analyses. Also explain the significance of the scores. P values should be presented for CMS and iSAFE scores, along with adjustment for multiple testing to ensure true significance. Only significant findings should be presented. This could remove some of the multiple hits that are not shared by European and Asian populations. In Table S3 indicate if P values are being presented or something else.

We added the following sentences to explain the use of both CMS and iSAFE and their scores in our analyses. “CMS comprehensively identifies sites that are most likely to have undergone a positive selective sweep, reporting a composite probability score from multiple selection tests for a given SNP¹⁹. iSAFE incorporates coalescent structures surrounding regions under selective sweep to further rank and pinpoint sites of positive selection²¹.” We report *P* values and adjust for multiple testing by calculating the false discovery rate (FDR, Benjamini-Hochberg) for each of the CMS scores that are now included in Table S1. We now report findings for SNPs that were significant (adjusted *P*<0.05) and are indicated as red dots in Figure 1b. iSAFE scores in Table S2 take into account *P* values whereby iSAFE>0.1 are significant with an empirical $P < 1 \times 10^{-4}$ (Akbari et al., *Nat. Methods* 2018) and hence are not reported as a separate column. Significant iSAFE SNPs are also labelled in red (Figure 1c). Orange dots indicate SNPs $0.095 < \text{iSAFE score} < 0.10$. As a result of this new analysis, no CMS SNPs in YRI were significant after FDR correction and also when using iSAFE. Hence, we removed the YRI data in Figure 1 and moved the data to Table S1. However, after correcting for FDR in CMS and with the overlap with iSAFE, we now present findings for rs12022319, rs4845490, rs4845491, and rs3737861 in CEU that comprises the positively selected CEU-*LCE1A-SMCP* haplotype. The CEU-*IVL* haplotype with $0.095 < \text{iSAFE score} < 0.10$ was found at the right shoulder of CEU-*LCE1A-SMCP* for which the functional significance for positive selection was further investigated. CMS signals in JPT/CHB in this region were no longer significant after FDR correction (*P* = 0.12) but new signals using iSAFE were found in JPT/CHB between *LCE1F* and *LCE1B*.

3. It could be hyperbole to state “As SMCP encodes a sperm mitochondrial cysteine-rich protein, the finding suggests a role for the positively selected SNPs in sperm function”

since the variants under selection in that region may be acting on other nearby genes. It is un-necessary for the current study and this sentence should be deleted.

This statement has been removed.

4. Some of the writing is overly complex: For example Figure 2 legend: “share commonality for both” – wouldn’t be more straightforward to say “both carry”? and later on: “A direct and positive correlation between the frequency of rs4845327-G, a tagging SNP for the CEU-IVL haplotype, and latitude reveals a selective advantage for CEU-IVL with increasingly Northern latitudes” – delete “and latitude” since this is what the authors then say is correlated with the tagging SNP.

We made the changes to Figure 2 legend.

5. Page 6. “The CEU-IVL haplotype contains expression quantitative trait loci (eQTLs)” is not entirely correct. It contains alleles of eQTLs. Similarly lower down (eQTLs associated with decreased expression...” and also on page 11.

We extensively revised this paragraph and clarify that the SNPs are annotated GTEx eQTLs that were found in these haplotypes.

6. Page 6. “is located 923 kb from the most 5’ EDC gene..” Surely EDC genes are not all in the same 5’ to 3’ direction – instead refer to proximal or distal chromosomal location.

We made the change to clarify that the 923 enhancer is located 923 kb from the most 5’ EDC gene *S100a10*.

7. Figure 3b shows the location of the orthologous deletion of the enhancer in mouse – however, two different deletions were generated. Show both and explain how these relate to the earlier study of Oh et al (2014).

We now show both deletion alleles (923^{del} and 923^{large}) in Figure 3a and 3b. We illustrate what was deleted in the 923^{del} allele (orange bar) and the 923^{large} allele (blue bar) and their alignments with respect to the human reference in Figure 3a. We also annotate the original conserved noncoding element (CNE) 923 as a gray bar with PhastCons blocks previously described by Oh et al. 2014. Guided by the strong H3K27ac epigenetic mark in NHEK that spans across the CNE 923 and PAM site inclusions for Cas9 cleavage, we deleted the orthologous enhancer region in the mouse that includes the CNE 923 and was accomplished in both 923^{del} and 923^{large} alleles. The mouse genomic region exists in the opposite orientation to that of the human. We made these clarifications in the manuscript text and Figure 3 legend.

8. Figure 3g. For the FCs for qRT-PCR, state what the mutant mice transcripts were compared with (wt mice?).

Mutant mice transcripts were all compared to WT mice. We indicate the change as $-\log_2$ (Fold Change from WT) on the y axis and in the figure legend in Figure 3d (originally Figure 3g).

9. Page 8. Presumably the lower level of B6 SNP alleles corresponds to mRNA – however as written this isn’t clear.

We clarified that the lower proportion of B6 SNP alleles were observed in *Lce6a* transcripts and included the change in the paragraph.

10. It isn't clear what the ATAC-Seq studies contribute to the current study, other than showing that the 923 enhancer indeed has this activity, since the regulatory elements do not seem to reside around IVL.

Our ATAC-seq studies identifies a role for the enhancer to modulate chromatin accessibility that was localized to the EDC. We found that three out of the 6 differentially accessible regions (DARs) discovered in both 923^{del/del} and 923^{large/large} keratinocytes compared to WT were exclusive to the EDC locus. Two DARs were less accessible and one DAR was more accessible. Although nearby gene expression levels were not affected by the DARs, the ATAC-seq findings demonstrate that deletion of the 923 enhancer impacts local EDC chromatin accessibility. We condensed these findings in the manuscript text and moved the original Figure 4 to Supplemental Figure 4 given the space limitations.

11. Page 10. When discussing luciferase assays, provide P values obtained when comparing activities of the different enhancers. Describe the SNP alleles that were used in the different constructs. Were any SNP alleles residing the different constructs identified from the selection studies described at the beginning of the manuscript. These data can sometimes be extracted from the manuscript, but they need to be presented in a more palatable form. Describing this clearly would certainly add to the value of the manuscript since these would be candidates within the enhancer that specifically alter IVL expression. Are any transcription factors predicted to map to these SNP alleles?

P values for the luciferase assays results have been added to the manuscript text. The CEU, JPT/CHB, and YRI alleles for the enhancer, promoter, and intron and the SNPs are shown in Fig 4a. We state that enhancer rs4845327, promoter rs1854779, intron rs7539232, and intron rs11205132 for CEU-IVL were found by iSAFE and are also IVL eQTL alleles specific to CEU-IVL (See also Table S4). We indicate these SNPs in orange bold in Figure 4a. We performed transcription factor binding analyses for all reference and alternate SNP alleles shown in Figure 4a and listed in Table S14. Only four SNPs resulted in differential TF binding for their alleles as shown in Figure 4c. We predict ZNF263 binding for enhancer rs1974141-A (loss of MAZ binding for G allele), IRF1 binding for iSAFE rs4845327-G (loss of SOX10 binding for G allele), NFIC binding for iSAFE promoter rs1854779-T (loss of SPI1 binding for C allele), and RELA binding for rs7517189-C (loss of GATA3 binding for G allele). Two of these SNPs included the positively selected enhancer rs4845327-G and promoter rs1854779-C identified by iSAFE. We further confirm preferential ZNF263, MAZ, NFIC and SPI1 transcription factor binding at polymorphic sites using motif and allele-specific analyses on ENCODE CHIP data that were only publicly available for these transcription factors (Table S15).

12. With the sentence: "Targeted sequencing of *Ivl* and *Lce6a* cDNAs obtained from 923delB6/BALB and 923largeB6/BALB heterozygous mouse skins revealed a significantly lower proportion of B6 SNPs from the 923del and 923large alleles for *Ivl* ..." (again use "skin" and not "skins"). Don't the authors mean B6 SNP alleles rather than SNPs.

We changed the sentence to indicate the use of two informative SNPs to determine B6 vs. BALB alleles.

13. The paragraph at the end of page 8 and beginning of page 9 makes no sense – please make clearer, explaining the rationale for the approach.

This allele-specific paragraph has been updated for clarity. “We next determined if the 923 enhancer regulates gene expression in *cis*. To do this, we performed allele-specific gene expression assays for *Ivl* and *Lce6a* in hybrid mouse skin. Allele-specific *Ivl* and *Lce6a* transcripts were distinguished using two informative SNPs for either the B6 or BALB/cBYJ (or BALB) allele in hybrid B6;BALB mouse tissue. Targeted sequencing of *Ivl* cDNA from 923^{del}B6/BALB and 923^{large}B6/BALB hybrid mouse skin revealed a significantly lower proportion of B6 transcripts from 923^{del} and 923^{large} alleles (13%) compared to the proportion of B6 transcripts from the WT allele observed in hybrid control B6/BALB mice (50%) (Fig. 3e) (ANOVA, Tukey post-hoc, $P < 0.01$). However, a significantly lower proportion of *Lce6a* B6 transcripts (3%) was observed in 923^{large}B6/BALB hybrid mouse skin only compared to 42% observed in hybrid control B6/BALB skin (ANOVA, Tukey post-hoc, $P < 0.01$). This further supports the hypothesis that the loss of a regulatory enhancer in the 923^{large} allele regulates allele-specific *Lce6a* expression. Together, our genetic findings identify *cis* regulation by the 923 enhancer for *Ivl*, thus establishing a 923 enhancer;*Ivl* regulatory module for the epidermis.”

14. Page 10: “Our discoveries for the positively selected IVL human haplotypes associated with differential IVL expressions...: change expressions to expression. Is there conservation between humans and mice at loci where variant alleles affect IVL expression from the 923 enhancer?”

We changed the sentence to “expression”. We now show a mouse alignment track to the human reference sequence of the 923 enhancer and *IVL* to indicate regions of mouse and human conservation on Figures 3a and 4a. For the enhancer locus, we further show mouse-human conservation at rs16834746-T and rs4845327-T that comprise variant alleles affecting IVL expression. rs1974141 is not found in the mouse as the mouse sequence is absent whereas rs12036697 differs from the mouse reference. We further show human-mouse conservation in the promoter at rs4845327. By contrast, of the 8 SNPs in the intron, 3 are not found in the mouse (no sequence), 3 are conserved, and the remaining 2 differ from the mouse. Figure 4a has been updated to visualize these findings.

15. Excerpt from 15 (and also 16-19 comments that are not shown here). The hypothesis that variants affecting IVL expression are modifiers of LOF FLG mutations is intriguing but confirmation of this will require much larger cohorts.

We deleted the section for our genetic AD findings since we cannot obtain a larger cohort at this time and given the previously raised concern that it detracts from the overall message. We are grateful for the comments (points 15 – 19) that are relevant to this section and will use them for a future manuscript submission.

20. In the discussion the authors mention copy number polymorphism of the *IVL* gene. Could this account for some of the different levels of expression associated with different haplotypes?

This copy number polymorphism (number of repeats) of the *IVL* gene does not appear to account for the different levels of expression for the different haplotypes. Both CEU and YRI haplotypes are commonly comprised of 8 “late” repeats each. “Late” indicates a segment of the *IVL* gene for which these repeats arose only in humans. Thus, it is unlikely that the CEU haplotype associated with increased *IVL* expression is explained by differences in the number of late repeats in contrast to YRI.

Reviewer #3 (Remarks to the Author):

1. Whilst this is of interest to those studying this genetic locus it is not particularly novel since there others have previously reported evidence of selection acting upon enhancer elements and the ‘923’ element has previously been reported to be an enhancer.

Thank you for the insightful comments. We emphasize the novelty of our work where we find positive selection acting upon an enhancer that to our knowledge has not been reported to date for human skin barrier evolution. We also emphasize the novelty of our work that reveals a requirement for enhancer to regulate gene expression in *cis* using knockout mice that also has been not demonstrated *in vivo* for the epidermis.

2. The population-based data in support of selection affecting the EDC locus is derived from publicly available datasets. It was not clear to me how many individuals were included in these datasets, moreover whilst it might be beyond the remit of this study it would be nice to see an independent validation cohort.

CMS scores were determined on HapMapII SNPs from 90 individuals for each of the 3 populations that were publicly available datasets at the time of the method development and analyses. We provided this detail in the Methods section. We sought to independently validate the significance of the positively selected CEU-*IVL* haplotype out-of-Africa. We utilized six publicly available genetic datasets from Finnish (<http://sisuproject.fi>; SISu v4.1, Sept. 2020 accessed), Great Britain (Sudlow et al., *PLoS Medicine* 2015), Estonia (Tasa et al., *EJHG* 2019), Denmark (Marett et al., *Nature* 2017), northern Sweden (Rentoft et al., *PLoS One* 2019), and the Netherlands (Boomsma et al., *Eur J Hum Genet*, 2014). We did not identify a significant correlation for the allele frequency for CEU-*IVL* coding SNP, rs2229496-A, with latitude ($R^2=0.04$; $p = 0.7$) and likely owing to the limitations of this dataset as the populations clustered between 54°-63° N. latitude and the allele frequencies ranged 0.88-0.97. We next attempted to include additional populations datasets between 17° and 40° N. latitudes using the European Genome-phenome Archive (EGPA) database. However, although datasets within these latitudes were available (Greece, Lebanon, Yemen and Chad), they were restricted against the use for population genetic studies. See also response to R4 point#5.

3. It would be interesting to see an analysis of conserved sequence elements and transcription factor binding sites in this enhancer element, perhaps along with some experiments (e.g. chromatin immunoprecipitation) to establish which transcription factors

are binding.

We now include a table of transcription factors that are predicted to bind to these SNP alleles (Table S14). We predict ZNF263 binding for enhancer rs1974141-A (loss of MAZ binding for G allele), IRF1 binding for iSAFE rs4845327-G (loss of SOX10 binding for G allele), NFIC binding for iSAFE promoter rs1854779-T (loss of SPI1 binding for C allele), and RELA binding for rs7517189-C (loss of GATA3 binding for G allele). See also Reviewer #1, point 11. Only four SNPs resulted in differential TF motifs *in silico* for their alleles as shown in Figure 4c. Two of these SNPs included the positively selected enhancer rs4845327-G and promoter rs1854779-C identified by iSAFE. We further confirm preferential ZNF263, MAZ, NFIC and SPI1 transcription factor binding at polymorphic sites using motif and allele-specific analyses on for cell lines and transcription factors where both ENCODE ChIP-seq and whole genome sequencing data were publicly available (Table S15, Fig S8).

Reviewer #4 (Remarks to the Author):

1. The selection tests: - There are, at this point, dozens of different tests of selection, each with their own sensitivity and accuracy for detecting different types of selective signatures. I think that CMS is a perfectly fine test to use, but it is important to contextualize exactly what it is measuring and what kind of sweep that this test more likely to find (Complete or incomplete, soft or hard, old or recent?)

Thank you for the helpful comments. We provide details to clarify the use of CMS that comprehensively integrates multiple statistical tests (iHS and XP-EHH for haplotype length; F_{ST} for population differentiation; ΔDAF for derived allele frequencies; and ΔiHH for absolute haplotype length) to identify genomic regions and the causative variants that are more likely to have undergone positive selection. The paper did not comment on the kind of sweep thus we refrain from commenting here. However, the study was powered to detect candidate causal variants for selective sweeps with frequencies as low as 20% yet with higher power for frequencies $>50\%$.

2. It is not clear to me if iSAFE is the right test here. I think this particular algorithm is designed to narrow down specific casual variants, rather than finding new selective signatures. In any case, the reason why these particular tests are used and what is expected to be found should be clarified. This is particularly important given the complex evolutionary history of the EDC.

We now provide details to justify why these tests are used (see also R1 point#2). CMS identifies sites that are most likely to have undergone a positive selective sweep, by incorporating multiple selection tests with a composite probability score for a given SNP¹⁹. iSAFE incorporates coalescent structures surrounding regions under selective sweep and was utilized here to further validate, rank, and localize sites of positive selection²¹. The use of iSAFE led us to identify the 4 SNPs for positive selection of *IVL*.

3. I am not sure how the significance of selection is calculated for these tests. It is, I think, important to know whether the values that they are observing for CMS are unusual just for the EDC locus or the whole genome? If they conduct this test, for example, to the LCT locus in CEU or EDAR locus in CHB, what would be the values? In fact, a desirable approach would be to identify the putatively selected haplotypes and using

either empirical and/or simulation-based methods to calculate the coalescent, date of the onset of the selection, and the selection coefficient. I know that this may be a little much to ask. However, I think the timing and nature of the adaptation of this locus would be essential to construct the eventual model of this locus' evolution.

The comment for significance was also raised by reviewer 1 (see #2). We determined the significance of selection for each EDC SNP in the EDC in the following manner. CMS SNP scores were genome-normalized for each population with subsequent identification of significant SNPs (FDR<0.05). We further validated the CMS SNPs using iSAFE statistical measures with scores <0.1 with $P < 1 \times 10^{-4}$. In doing so, we identified multiple signals of positive selection in the EDC in JPT/CHB and CEU. Here we conducted the same statistical analyses for *EDAR* SNPs in JPT/CHB to demonstrate the validity of our approach. We also found significance for 51 *EDAR* SNPs with CMS scores>10 and $P < 0.05$ including *EDARV370A* variant, rs3827760, that exhibited the lowest $P = 3.9 \times 10^{-12}$ and was previously reported by Grossman et al. and Kamberov et al. to be positively selected in Han Chinese (Grossman et al., Science 2010 and Kamberov et al., Cell 2013). Although the CMS scores were higher for *EDAR* in JPT/CHB than what we observed in the EDC, we nevertheless find positive selection and the significance for distinct SNPs and their haplotypes in the EDC. We focused on the EDC region that were motivated by our findings for recent evolution in the EDC in mammalian phylogeny (Goodwin and de Guzman Strong, Front Genet. 2017) and identification of the EDC as the most rapidly diverging gene cluster in the human-chimp genome comparison (Mikkelsen et al., Nature 2005). We agree that it is important to determine the timing and nature of the adaptive allele but the approach using empirical and/or simulation-based methods is beyond the scope of this study as mentioned.

4. Enhancer work:

- It is clear both from existing work as well as the authors' mice work that the region of interest is an enhancer. What is more interesting to me that their top SNP actually is hitting this enhancer sequence. This should be stated in the text and Figure 3A. I am disappointed that the authors did not reconstruct the allelic variation in the mouse, which may provide a model for evolutionary work as well as AD susceptibility.

We believe the reviewer is referring to rs4845327 identified by iSAFE that is located in the enhancer. rs4845327-G variant found in the positively selected CEU-*IVL* is associated with increased *IVL* and reporter activity (Fig2, Fig4). We state this in the text and highlight its location in Fig. 2a (bolded lollipop) and Fig. 4a (orange). We also included iSAFE promoter SNP, rs1854779, for which the T allele is also associated with increased *IVL* and reporter activity and amended Figure 4a to show the location of rs1854779. We further report prediction for IRF1 transcription factor binding to rs4845327-G (in contrast to SOX10) and NFIC to rs1854779-T (in contrast to SPI1). The reconstruction of allelic variation in the mouse is beyond the scope of this report but we are grateful for the reviewer's comment and aim to pursue this in future studies. However, we address the significance of preferential transcription factor binding using ENCODE ChIP and motif and enrichment analyses.

5. Modeling the evolution and functional relevance of this locus:

- I am very skeptical about the latitude connection. Any variant that have substantially different allele frequencies in Africa and Eurasia will appear to be correlated with latitude. I think, larger, within-continent datasets is required for a better understanding of this issue.

We further examined positive selection associated with latitude using within-continent datasets for both Europe and Africa. We included 6 additional within-continent datasets in Europe that were publicly available. We included European datasets for Finnish ([URL:http://sisuproject.fi]; SISu v4.1, date (Sept, 2020) accessed), Great Britain (Sudlow et al., PLoS Medicine 2015), Estonia (Tasa et al., EJHG 2019), Denmark (Maretty et al., Nature 2017), northern Sweden (Rentoft et al., PLoS One 2019), and the Netherlands (Boomsma et al., Eur J Hum Genet, 2014). We did not identify a significant correlation for the allele frequency for CEU-IVL coding SNP, rs2229496-A, with latitude ($R^2=0.04$; $\rho = 0.7$). We attempted to include additional populations datasets between 17° and 40° N. latitudes using the European Genome-phenome Archive (EGPA) database. However, although datasets within these latitudes were available (Greece, Lebanon, Yemen and Chad), they were restricted against the use for population genetic studies. We performed the same analyses for the 1KGP datasets in Africa and found weak positive correlation

that was not significant ($R^2=0.1$; $\rho = 0.95$). Given these findings and the limitations of these datasets, we revised our claim to indicate that the CMS and iSAFE positively selected signals are associated with out-of-Africa using latitude as a proxy and not as a direct determinant of positive selective pressure which would require further testing. This finding is in contrast to the EDC SNP, rs2711, that is globally common and not correlated with latitude ($R^2= 0.55$; $\rho = 0.0086$; Figure S1). As more genomic datasets become available, we will revisit this question in the near future.

6. I made a quick analysis (in PMID: 31217584 data) and it seems that the putatively selected SNP rs1854779 is very significantly associated with White Blood Cell Counts. How can the authors dismiss immune-mediated phenotypes as a potential adaptive force in shaping the variation in this locus?

We looked at the data reported in Wojcik et al. PAGE study (PMID: 31217584). We could not find evidence for rs1854779 to be significantly associated with White Blood Cell Counts in their study. We looked further and also did not find associations for any of our positively selected SNPs with any other phenotypic traits in the PAGE study.

7. I think there is a lack of specifics when it comes to what exactly the putatively selected haplotype(s) does other than changing expression levels. Moreover, its connection to skin disease has barely been discussed. Even if it is the case that this variation is linked to Vitamin D deficiency, how is lower expression of IVL help with that (weakened Epidermal Layer and more UV penetration?). How is that linked to AD - is weakened epidermal barrier function a hallmark of AD?

We added the following sentences in the discussion to address the comments above with the caveat that we removed the AD genetics section per R1 comments. “Selective sweeps for haplotypes, CEU-LCE1A-SMCP with increased HRNR and CEU-IVL with increased IVL expressions, suggest a benefit for higher protein dosage for the evolving skin barrier. This notion is supported by 1) the absence of common deleterious (albeit truncating) variants in strong linkage within these haplotypes and 2) the highly repetitive nature of HRNR and IVL that lengthens the protein structures observed across mammalian and primate phylogenies⁴⁴⁻⁵² and anticipated trajectories in humans. The functional benefits for increased HRNR and IVL have yet to be functionally determined yet it can be speculated that protein dosage modulation provides an innovative strategy to calibrate skin barrier function, i.e. permeability, to the environment. Indeed, weakened epidermal barrier function is a hallmark of atopic dermatitis, a common inflammatory disease, owing to the discovery for >100 loss-of-function variants for the highly repetitive FLG gene that are population-specific, and an increase in prevalence worldwide⁶¹.”

8. Some minor suggestions for clarity and organization:

- What is the most 5' EDC gene (line 142)?

S100a10 is the 5' most EDC gene. We made this change in the text.

9. The luciferase assay and the reason as to why these experiments are conducted should be more clearly explained in my opinion, especially within the context of the evolution of the variation in this locus.

- Too much going on in figure 3 and they are not discussed in the main text. The figure can be simplified, or some parts can be put in supplementary

We revised this section with the following sentences to justify the use of luciferase assays for positive selection of CEU-*IVL*. Luciferase assays were conducted to better understand the regulatory activity of the 923 enhancer:*IVL* promoter ± enhancer of the positively selected CEU-*IVL* haplotype associated with increased *IVL* expression. With new knowledge for the 923 enhancer:*IVL* regulatory module in mice, we next sought to determine the mechanism by which increased *IVL* expression is associated with the positively selected CEU-*IVL* human haplotype. We hypothesized increased regulatory activity by the CMS/iSAFE SNPs in the orthologous 923 enhancer and promoter for CEU-*IVL*. We performed luciferase assays on population-specific alleles for the *IVL* promoter, first (noncoding) exon, and intron, of known regulatory activity²⁹ to determine the effects in CEU-*IVL* in comparison to JPT and YRI common haplotypes. We simplified Figure 3 and removed the original panels for c, e, and f and included only e and f panels as Supplemental Figure 5 instead.

Reviewers' Comments:

Reviewer #1:

Remarks to the Author:

Understanding the molecular evolution of the Epidermal Differentiation Complex (EDC) locus is of great interest in understanding the role of the skin barrier in health and disease. The authors have done an excellent and thorough job of revising this manuscript and provide a much more coherent and concise report. The results are now clearly presented and are both intriguing and compelling and provide important information on the evolution of this region.

Most of my comments are minor and are a request for more clarity or are grammar corrections:

I have issues with the title which seems confusing – can the authors provide something clearer – “enhancer:involucrin regulatory module allele” makes no sense.

Page 5: “The HRNR-CRNN signals span (“across” is not needed here) multiple genes...

And lower down “span (“across” not needed here).

Page 6, CEU-IVL haplotype... Paragraph – line three:

“Of the four CMS/iSAFE SNPs “ON THE “ is preferable to “for”.

Line 5: “were” not “was”.

Line 10: “expression” not “expressions” “of” (not “for”.

Page 7:

I am not clear why the authors have continued to speak about the most 5’ EDC gene – since genes in the EDC are not all expressed from the same strand. I again ask, should this be telomeric or centromeric?

Later on in this paragraph the authors talk about the H3K27Ac epigenetic mark, but don’t state where it resides with respect to the 923 element.

Reviewer #3:

Remarks to the Author:

I am satisfied with the additional in silico analysis of transcription factor binding motifs. Given that 6 additional genetic datasets are available (Finnish, Great Britain, Estonia, Denmark, Sweden, Netherlands) it would be valuable to see an analysis of these displayed in Figure 1 to confirm the presence of selection acting on the EDC.

Reviewer #4:

Remarks to the Author:

I thank the authors for their responses to my comments. I especially appreciate the willingness to consider the issue with the latitude correlation (or lack thereof). Having said that, I still have two concerns. One is that the authors were not able to show that the functional effect of the allelic variation in a direct experiment. At the very least, this should be clearly stated. Second, rs1854779 is clearly associated with White Blood Count (rs1854779) - Please do check GWAS ATLAS - Phewas analysis (<https://atlas.ctglab.nl>)- There seems to be a pretty strong association found only by looking into diverse populations. The association is reported by <https://pubmed.ncbi.nlm.nih.gov/31217584/>. This should be discussed in my opinion.

Reviewers' comments:

We thank the reviewers for their helpful comments. Specific changes to the manuscript and supplemental information are indicated as bars on the left margins and underlined in the text where possible.

Reviewer #1 (Remarks to the Author):

1. Understanding the molecular evolution of the Epidermal Differentiation Complex (EDC) locus is of great interest in understanding the role of the skin barrier in health and disease. The authors have done an excellent and thorough job of revising this manuscript and provide a much more coherent and concise report. The results are now clearly presented and are both intriguing and compelling and provide important information on the evolution of this region.

Thank you for your enthusiasm and comments to improve our manuscript.

2. Most of my comments are minor and are a request for more clarity or are grammar corrections: I have issues with the title which seems confusing – can the authors provide something clearer – “enhancer:involucrin regulatory module allele” makes no sense.

We removed “regulatory module” in the title. The title now reads “Selective sweep for an enhancer:involucrin allele identifies skin barrier adaptation out-of-Africa”.

3. Page 5: “The HRNR-CRNN signals span (“across” is not needed here) multiple genes... And lower down “span (“across” not needed here).

We removed “across” in both sentences on page 5.

4. Page 6, CEU-IVL haplotype.... Paragraph – line three: “Of the four CMS/iSAFE SNPs “ON THE “ is preferable to “for”.
Line 5: “were” not “was”.
Line 10: “expression” not “expressions” “of” (not “for”).

We made the specific changes to the sentences on page 6.

5. Page 7: I am not clear why the authors have continued to speak about the most 5' EDC gene – since genes in the EDC are not all expressed from the same strand. I again ask, should this be telomeric or centromeric?

We removed “5'” and changed it to centromeric.

6. Later on in this paragraph the authors talk about the H3K27Ac epigenetic mark, but don't state where it resides with respect to the 923 element.

We apologize for the confusion. We clarified this issue with the following sentences in the paragraph. “Our strategy aimed to delete the conserved non-coding element 923 with known epidermis-specific regulatory activity^{7,25} and epigenetically marked by H3K27ac histone acetylation observed in ENCODE human keratinocytes. We targeted a larger 934 enhancer region given the H3K27ac epigenetic mark.”

Reviewer #3 (Remarks to the Author):

I am satisfied with the additional in silico analysis of transcription factor binding motifs. Given that 6 additional genetic datasets are available (Finnish, Great Britain, Estonia, Denmark, Sweden, Netherlands) it would be valuable to see an analysis of these displayed in Figure 1 to confirm the presence of selection acting on the EDC.

Thank you for your helpful comments. We sought to perform positive selection analyses on these 6 additional genetic datasets. Unfortunately, our positive selection analyses require individual genotype files that are not publicly available for these datasets. Thus, we performed iSAFE analyses on two additional 1KGP European populations (FIN and IBS). Here we confirm positive selection for the same *LCE1B-IVL* region and *LCE1A-SMCP* and *IVL* haplotypes in FIN and IBS as shown in Figure S1 and Table S3. We modified the following sentence to report this finding.

Page 5 - "iSAFE provided higher resolution and further validation of CMS signal surrounding the *LCE1B-SMCP* region and elucidation of the positively selected *IVL* shouldering region in CEU that we also found to be positively selected in European 1KGP populations, FIN (Finnish in Finland) and IBS (Iberian population in Spain) (Table S2, Fig. S1)."

We also added the following sentences.

Page 6, Results – "The same haplotypes were also found in FIN and IBS (Table S3)."

Page 13, Discussion – "Interestingly, the backbone of the CEU-*IVL* haplotype appears to have emerged in Africa and rose to near-fixation frequency in CEU and other European populations."

Reviewer #4 (Remarks to the Author):

1. I thank the authors for their responses to my comments. I especially appreciate the willingness to consider the issue with the latitude correlation (or lack thereof).

Thank you for your helpful comments and enthusiasm for our manuscript.

2. Having said that, I still have two concerns. One is that the authors were not able to show that the functional effect of the allelic variation in a direct experiment. At the very least, this should be clearly stated. Second, rs1854779 is clearly associated with White Blood Count (rs1854779) - Please do check GWAS ATLAS - Phewas analysis (<https://atlas.ctglab.nl>)- There seems to be a pretty strong association found only by looking into diverse populations. The association is reported by <https://pubmed.ncbi.nlm.nih.gov/31217584/>. This should be discussed in my opinion.

Thank you for these important points. We agree and add the following statements in the 3rd paragraph in the Discussion. "Future experiments are needed to isolate and further determine the functional effects of each SNP on *IVL* expression and even on other cell types. rs1854779 was recently determined to be associated with white blood cell count and was discovered by including diverse human populations to comprehensively examine phenotypic traits³⁷. This, together with our human evolutionary findings, generates a hypothesis for a human skin diversity SNP to rapidly evolve and modulate the functional interface of the skin and immune systems."